# Dynamic Na$^+$/H$^+$ exchanger 1 (NHE1) – calmodulin complexes of varying stoichiometry and structure regulate Ca$^{2+}$-dependent NHE1 activation

Lise M Sjøgaard-Frich[1†], Andreas Prestel[2†], Emilie S Pedersen[2], Marc Severin[1], Kristian Kølby Kristensen[3,4], Johan G Olsen[2], Birthe B Kragelund[2]*, Stine Falsig Pedersen[1]*

[1]Section for Cell Biology and Physiology, Department of Biology, Faculty of Science, University of Copenhagen, Copenhagen, Denmark; [2]Structural Biology and NMR Laboratory, Department of Biology, Faculty of Science, University of Copenhagen, Copenhagen, Denmark; [3]Finsen Laboratory, Rigshospitalet, Copenhagen, Denmark; [4]Biotech Research and Innovation Centre (BRIC), University of Copenhagen, Copenhagen, Denmark

*For correspondence:
bbk@bio.ku.dk (BBK);
sfpedersen@bio.ku.dk (SFP)

†These authors contributed equally to this work

Competing interests: The authors declare that no competing interests exist.

**Abstract** Calmodulin (CaM) engages in Ca$^{2+}$-dependent interactions with numerous proteins, including a still incompletely understood physical and functional interaction with the human Na$^+$/H$^+$-exchanger NHE1. Using nuclear magnetic resonance (NMR) spectroscopy, isothermal titration calorimetry, and fibroblasts stably expressing wildtype and mutant NHE1, we discovered multiple accessible states of this functionally important complex existing in different NHE1:CaM stoichiometries and structures. We determined the NMR solution structure of a ternary complex in which CaM links two NHE1 cytosolic tails. *In vitro*, stoichiometries and affinities could be tuned by variations in NHE1:CaM ratio and calcium ([Ca$^{2+}$]) and by phosphorylation of S648 in the first CaM-binding α-helix. In cells, Ca$^{2+}$-CaM-induced NHE1 activity was reduced by mimicking S648 phosphorylation and by mutation of the first CaM-binding α-helix, whereas it was unaffected by inhibition of Akt, one of several kinases phosphorylating S648. Our results demonstrate a diversity of NHE1:CaM interaction modes and suggest that CaM may contribute to NHE1 dimerization and thereby augment NHE1 regulation. We propose that a similar structural diversity is of relevance to many other CaM complexes.

## Introduction

Calmodulin (CaM) is a ubiquitously expressed EF-hand Ca$^{2+}$-binding hub protein, which regulates a plethora of Ca$^{2+}$-dependent cellular processes such as ion transport, muscle contraction, proliferation, and apoptosis (*Berchtold and Villalobo, 2014*; *Chin and Means, 2000*; *Villalobo et al., 2018*; *Villarroel et al., 2014*; *Yap et al., 2000*). CaM is a small, α-helical protein consisting of two similar, but not identical, N- and C-terminal domains (for CaM termed the N-lobe and C-lobe, respectively) connected by a helical linker with a disordered, flexible hinge. Upon changes in the free cytosolic Ca$^{2+}$ concentration, [Ca$^{2+}$]$_i$, CaM binds up to four Ca$^{2+}$ ions. This induces structural rearrangements exposing hydrophobic patches through which CaM interacts with numerous structurally and functionally different proteins (*Tidow and Nissen, 2013*; *Villalobo et al., 2018*). In most cases where this has been studied to date, CaM wraps both its lobes around its α-helix-forming targets, exemplified by the interaction with myosin light chain kinase (*Ikura et al., 1992*; *Meador et al., 1992*). However, recent structures have shown diversity in binding and a range of CaM binding modes and motifs exist. Indeed,

in some cases, CaM with $Ca^{2+}$ present only in one lobe is bound to the target, whereas in other cases only the apo-form of CaM is bound (*Lee et al., 2019*; *Nunomura et al., 2014*; *Nunomura et al., 2011*). Still, common features of CaM-binding regions, such as high α-helix propensity, net positive charge, and obligate hydrophobic docking residues, exist (*Villalobo et al., 2018*). CaM also engages with many membrane proteins and examples include the CaM substrates estrogen receptor alfa (ERα; *Li et al., 2005*; *Zhang et al., 2012*), the voltage-gated $K^+$ channel Kv10.1 (eag1; *Schönherr et al., 2000*; *Whicher and MacKinnon, 2016*), and aquaporin 0 (*Reichow et al., 2013*).

One widely studied CaM-binding membrane protein is the $Na^+/H^+$ exchanger, NHE1 (SLC9A1), a major acid-extruding transporter localized in the plasma membrane in essentially all mammalian cells studied (*Orlowski and Grinstein, 2011*; *Pedersen and Counillon, 2019*). Upon activation by cytosolic acidification or a wide range of mitogenic and other signaling events, NHE1 extrudes $H^+$ from the cytosol in exchange for $Na^+$ ions. In addition to its role in intracellular pH ($pH_i$) regulation, NHE1 regulates key cellular functions such as proliferation, growth/death balance, cell volume, cell motility, and tissue homeostasis (*Orlowski and Grinstein, 2011*; *Pedersen and Counillon, 2019*). Consequently, NHE1 hyperactivity is implicated in many important pathologies including cancers (*Cardone et al., 2005*; *Stock and Pedersen, 2017*), as well as cardiovascular disease, especially ischemia-reperfusion damage and cardiac hypertrophy (*Imahashi et al., 2007*; *Karmazyn et al., 1999*; *Nakamura et al., 2008*) and non-alcoholic steatohepatitis (*Prasad et al., 2014*).

NHE1 which, like many other membrane transport proteins, functions as a dimer (*Fafournoux et al., 1994*; *Pedersen and Counillon, 2019*), consists of a 12-transmembrane (TM)-helix transport domain and an ~300 residue long cytoplasmic C-terminal domain with extensive disordered regions (*Nørholm et al., 2011*). Via this tail, NHE1 activity is regulated through multiple phosphorylation–dephosphorylation events and interactions with a plethora of binding partners, one of which is CaM (for reviews, see *Hendus-Altenburger et al., 2014*; *Orlowski and Grinstein, 2011*; *Pedersen and Counillon, 2019*). CaM binds directly to the cytoplasmic domain of NHE1 in the presence of $Ca^{2+}$ (*Bertrand et al., 1994*; *Köster et al., 2011*) as originally demonstrated *in vitro* by binding of either an NHE1 fusion protein or full-length protein (*Bertrand et al., 1994*). The binding of $Ca^{2+}$-loaded CaM to NHE1 in response to an increase in $[Ca^{2+}]_i$ elicits an alkaline shift in the $pH_i$ sensitivity of NHE1 in cells, resulting in its activation at less acidic $pH_i$ (*Bertrand et al., 1994*; *Wakabayashi et al., 1994*). Mechanistically, NHE1:CaM interaction has been proposed to activate NHE1 by relieving an autoinhibitory interaction of the C-terminal tail of NHE1 with its TM region (*Ikeda et al., 1997*; *Wakabayashi et al., 1997*). CaM binding has also been implicated in activation of NHE1 by a wide range of hormones and growth factors, including epidermal growth factor, serotonin, angiotensin, and endothelin (*Coaxum et al., 2009*; *Li et al., 2013*; *Turner et al., 2007*). NHE1:CaM interaction was suggested to be regulated by Akt kinase-mediated phosphorylation of S648 of human NHE1. However, the role of S648 phosphorylation in NHE1 regulation has been reported both as inhibitory (*Snabaitis et al., 2008*) and stimulatory (*Meima et al., 2009*), pointing to a possible context dependence of NHE1:CaM complex formation.

Two neighboring α-helical CaM-binding sites of NHE1 each bind CaM, but with different affinities. For an NHE1 variant deleted in CaM-binding region 2 (CB2, defined as D656-L691), $K_d$ was ~20 nM and for an NHE1 variant deleted in CaM-binding region 1 (CB1, defined as N637-A656) $K_d$ was ~350 nM (*Figure 1a, b*, *Bertrand et al., 1994*). Testifying to its importance, deletion or mutation of CB1 resulted in loss of NHE1 $[Ca^{2+}]_i$ responsiveness (*Bertrand et al., 1994*; *Wakabayashi et al., 1994*). A crystal structure of CaM bound to a peptide of NHE1 containing both CaM-binding sites (A622-R690) revealed an unusual binding mode. CaM assumed an extended configuration, and the N-terminal CaM-binding site of NHE1 (H1 in the following) made an uncommon interaction with the 'back side' of the C-lobe, whereas the N-lobe of CaM engaged with the low-affinity C-terminal site of NHE1 (H2 in the following) forming an α-helix in the hydrophobic ligand-binding groove (*Köster et al., 2011*). This extraordinary mode of CaM binding, combined with the fact that NHE1 forms active dimers in the membrane, raises the possibility that in solution other types of CaM complexes of relevance to the function of NHE1 may exist. So far, the only analysis of the complex in solution was conducted using small-angle X-ray scattering (*Köster et al., 2011*). Here, the derived molecular envelope showed a poor fit to the crystal structure, further spurring a re-evaluation of the conformational landscape of the CaM:NHE1 complex to understand its functional relevance.

To explore the structural landscape of possible NHE1:CaM complexes in solution and elucidate the precise mechanism and role of phosphorylation of S648, which is localized in the C-terminal part of the

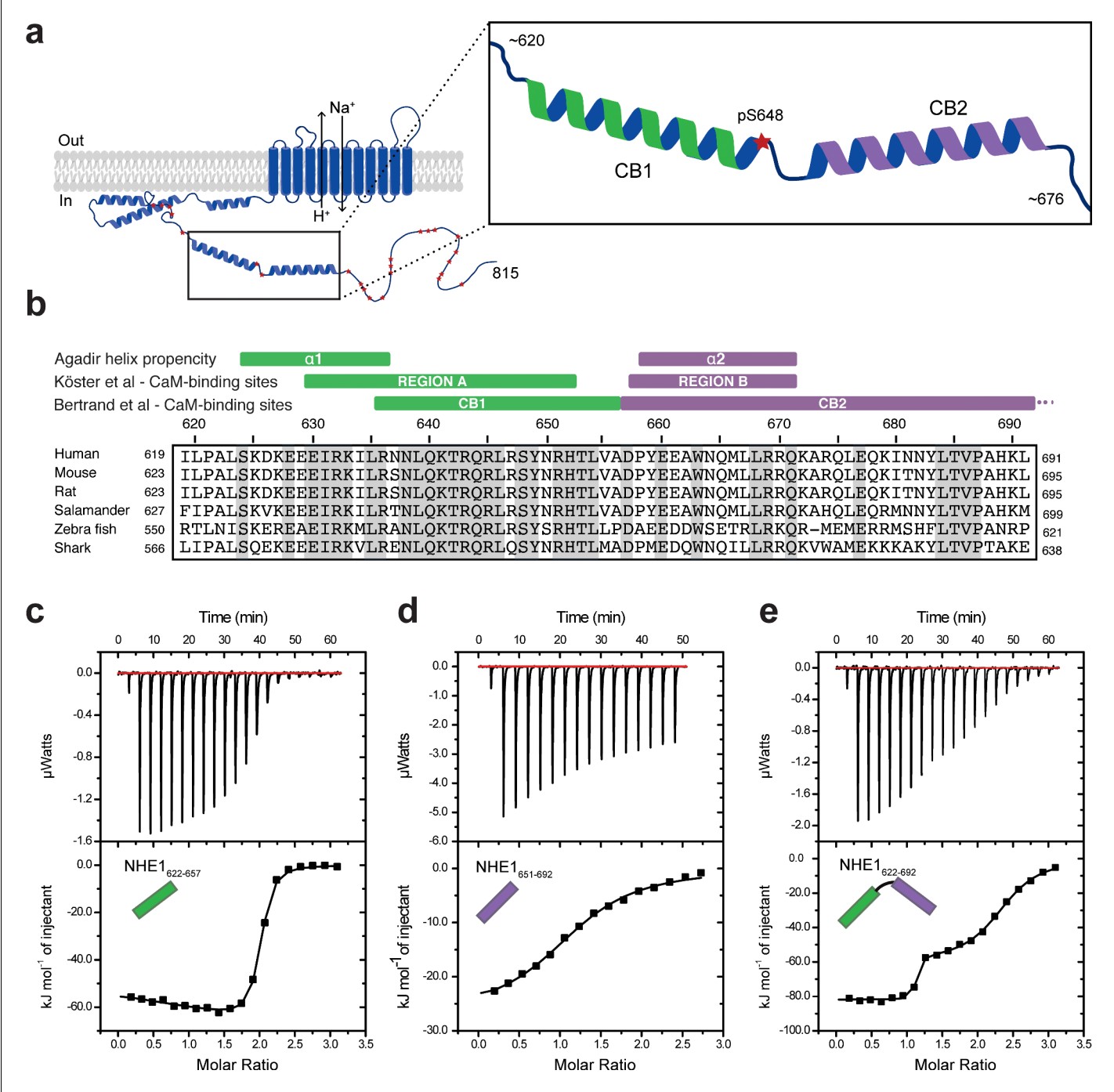

**Figure 1.** Na$^+$/H$^+$-exchanger:calmodulin (NHE1:CaM) interactions *in vitro* adopt different stoichiometries. (**a**) Sketch of NHE1 (blue) in the membrane highlighting helical elements in the cytoplasmic tail and known phosphorylation sites (red stars). Magnified region depicts the two previously described CaM-binding (CB) α-helices CB1 (green) and CB2 (purple) with the phosphorylated S648. (**b**) Alignment of the sequences of human, mouse, rat, salamander, zebra fish, and shark NHE1 is highlighted in gray identical residues. Indicated above are structural and functional regions based on either an Agadir α-helix prediction, the CaM-binding region A (green) and region B (purple), as defined by *Köster et al., 2011* or the CaM-binding region I (CB1) and II (CB2), as defined by *Bertrand et al., 1994*. (**c**)-(**e**). Binding NHE1 to CaM by isothermal titration calorimetry (ITC). In representative ITC experiments, we injected (**c**) NHE1$_{622-657}$ or (**d**) NHE1$_{651-692}$ or (**e**) NHE1$_{622-692}$ into CaM. The upper parts show baseline-corrected raw data from the titrations, and the lower parts the normalized integrated binding isotherms with the fitted binding curves assuming a single binding event in (**d**) and two binding events in (**c**) and (**e**). The NHE1 peptides titrated into CaM are shown in cartoon.

The online version of this article includes the following source data for figure 1:

**Source data 1.** Isothermal titration calorimetry raw data and fits of triplicate experiments for the titration of calmodulin with H1, H2, and H1H2 (relating to *Figure 1c–e*).

first CaM-binding helix of NHE1, we employed a combination of solution state-nuclear magnetic resonance (NMR) spectroscopy, isothermal titration calorimetry (ITC), NHE1 mutations, proximity ligation assays (PLA), and real-time analyses of NHE1-mediated acid extrusion after stable expression of wild-type (WT) and variant NHE1 in mammalian fibroblasts. Our findings reveal that, in solution, CaM interacts with NHE1 by exploiting different binding modes and stoichiometries, thereby forming several possible structures. Accordingly, in cells, $Ca^{2+}$-induced NHE1 activation, but not NHE1:CaM proximity, was modulated by mutations preventing (S648A) or mimicking (S648D) phosphorylation, as well as reduced by charge-reversal (CR) mutations of the first CaM-binding site. This was not dependent on Akt kinase, inhibition of which did not affect $Ca^{2+}$-induced activation of NHE1 under these conditions. Consistent with this notion, S648 phosphorylation could be mediated by other Ser/Thr kinases. We propose that the interaction between CaM and NHE1 is highly dynamic, involving both CaM-binding helices of NHE1, and with a functional role of S648 that is not exclusively downstream from Akt. We suggest that the high degree of diversity observed for NHE1:CaM complexes may be relevant to other CaM complexes, particularly those involving membrane proteins.

## Results

### CaM interacts with NHE1 in multiple conformations *in vitro*

In order to characterize the interaction of CaM with NHE1 *in vitro*, we designed three NHE1 peptides: human (h)NHE1$_{622-657}$ (H1), hNHE1$_{651-692}$ (H2), and hNHE1$_{622-692}$ (H1H2); all based on previous studies (*Köster et al., 2011*), sequence conservation, and helix propensity calculations (*Figure 1a, b*). First, ITC was applied to determine binding thermodynamics and stoichiometries of the NHE1-derived peptides and CaM. In line with previous results (*Bertrand et al., 1994*), H1 bound CaM with high affinity. In addition, we surprisingly observed that one CaM molecule bound two H1 peptides (n = 1.92 ± 0.06, *Figure 1c*). Fitting the ITC data to a binding model with two independent sites revealed similar high affinities ($K_{d,1}$ = 27 ± 8 nM; $K_{d,2}$ = 42 ± 6 nM, *Figure 1c*), but with different enthalpic and entropic contributions (*Table 1*). We then addressed the binding of the second helix, H2, to CaM. Here, the affinity was almost three magnitudes weaker ($K_d$ = 9.0 ± 0.6 µM) than that of H1. This interaction showed only one transition, indicating the formation of a 1:1 complex (n = 1.15 ± 0.08, *Figure 1d*). Finally, the binding isotherm for the H1H2 peptide displayed two distinct transitions, suggesting successive binding events and/or diverse binding modes with different CaM:H1H2 stoichiometries. Fitting to a two-site binding model allowed us to determine the affinity of the first binding event to $K_{d,1}$ = 0.27 ± 0.07 nM with a stoichiometry of 1 ($n_1$ = 1.04 ± 0.05), and the affinity of the second binding event to $K_{d,2}$ = 560 ± 15 nM with $n_2$ = 1.22 ± 0.06, summing up to a total stoichiometry of 1:2 (*Figure 1e*).

These results demonstrate the formation of variable NHE1:CaM complexes with different stoichiometries as well as thermodynamic profiles *in vitro*. Furthermore, it suggests that the interaction between NHE1 and CaM cannot be fully explained by the crystallized 1:1 complex alone.

### Structural insight into the NHE1 complexes with CaM in solution

We next set out to obtain structural insight into these additional complexes of CaM:H1H2 in solution using NMR spectroscopy, which is uniquely powerful to obtain information about the structure and dynamics of proteins and multi-state protein complexes (*Alderson and Kay, 2021*). The NMR signals of CaM in complex with H1H2 were at all tested stoichiometries and conditions severely broadened

**Table 1.** Thermodynamic parameters of NHE1:CaM interactions.

| | $\Delta H_1$ (kJ/mol) | $T\Delta S_1$ (kJ/mol) | $K_{d,1}$ (nM) | $n_1$ | $\Delta H_2$ (kJ/mol) | $T\Delta S_2$ (kJ/mol) | $K_{d,2}$ (nM) | $n_2$ |
|---|---|---|---|---|---|---|---|---|
| H1 (one site) | −60 ± 3 | −17 ± 2 | 23 ± 5 | 1.92 ± 0.06 | – | – | – | – |
| H1 (two site) | −39 ± 11 | 4 ± 10 | 27 ± 8 | 0.92 ± 0.05 | −83 ± 8 | −41 ± 8 | 42 ± 6 | 0.98 ± 0.01 |
| H2 (one site) | −29 ± 4 | 0 ± 3 | 8900 ± 600 | 1.15 ± 0.08 | – | – | – | – |
| H1H2 (two site) | −84 ± 5 | −29 ± 5 | 0.27 ± 0.07 | 1.04 ± 0.05 | −62 ± 3 | −26 ± 3 | 560 ± 10 | 1.22 ± 0.06 |
| H1H2 pS648 (two site) | −75 ± 3 | −27 ± 2 | 5 ± 4 | 0.93 ± 0.02 | −58 ± 7 | −26 ± 8 | 2700 ± 700 | 1.16 ± 0.06 |

NHE1: Na$^+$/H$^+$-exchanger; CaM: calmodulin.

and made direct determination of the three-dimensional structure of this complex by NMR impossible. We therefore addressed this structure using NMR data recorded on the CaM complexes with the individual H1 and H2 peptides. From a comparison of these high-quality assigned spectra to the broad signals in the CaM:H1H2 spectra, we could extract information and infer important structural details of the CaM:H1H2 complexes.

## Both lobes of CaM can interact independently with H1 with high affinity

En route to understanding the spectra of the CaM:H1H2 NMR spectra, we started out titrating $^{15}$N-labeled CaM with increasing concentrations of unlabeled H1 and followed the changes in shape, intensity, and position of the NMR peaks (*Figure 2a*). Upon addition of H1 (50 µM CaM + 50 µM NHE1), most signals from both CaM lobes broadened drastically, indicating exchange between free and bound state(s) on an intermediate NMR timescale ($k_{ex} \approx \Delta\delta$) (*Prestel et al., 2018*; *Figure 2b*). This suggests dynamics in the complex occurring on the microsecond to millisecond timescale. After addition of excess H1 peptide (50 µM CaM + 125 µM H1, denoted 1:2 in the following), the signals resharpened and no further spectral changes were observed upon further addition of H1 (*Figure 2a, b*). This is in line with the CaM:H1 stoichiometry of 1:2 obtained by ITC. Although at a molar ratio of 1:1 all the NMR signals were broadened by exchange processes, it was evident that signals from the C-lobe of CaM were located close to the position of the fully bound state (1:2), while signals of the N-lobe were positioned close to those of the free form of CaM, as exemplified for G135 and G62, respectively (*Figure 2b*). This suggested that in the main state at a 1:1 stoichiometry H1 is bound to the higher affinity C-lobe, as depicted in the cartoon in *Figure 2c*. Not all peaks could be followed throughout the titration, but the high-quality NMR spectra of the 1:2 (CaM:H1) complex allowed assignments of CaM in the ternary complex and mapping of the chemical shift perturbations (CSPs) onto the sequence (*Figure 2d*). Large CSPs were observed for residues throughout CaM, highlighting binding of H1 to both lobes. The rate constants for dissociation of H1 from the C-lobe and N-lobe were determined by 2D NMR lineshape analysis to $k_{off,1} = 40 \pm 5$ s$^{-1}$ and $k_{off,2} = 16 \pm 3$ s$^{-1}$, respectively (*Figure 2—figure supplement 1a*, *Table 2*), revealing a >10 times faster association with the C-lobe than with the N-lobe. To further substantiate these results, the opposite titration was performed: $^{15}$N-labeled H1 was monitored as the concentration of unlabeled CaM was increased. Only a few signals from the free peptide were visible at 37°C, presumably due to its disordered nature, while at a 1:2 (CaM:H1) molar ratio, two sets of signals became distinct and could be assigned to H1 bound to either of the two CaM lobes, respectively (*Figure 2—figure supplement 2a*). When more CaM was added, the set of signals originating from H1 bound to the N-lobe decreased in intensity and vanished at a molar ratio of 3:1 (CaM:H1). At this condition, the peptide was almost exclusively bound to the higher affinity C-lobe of CaM (*Figure 2—figure supplement 2a*).

We next titrated $^{15}$N-labeled CaM with the low-affinity H2 peptide. Here, in contrast to the H1 titration, the NMR signals were gradually moving with increasing peptide concentration, indicating the free and bound states to be in fast exchange on the NMR timescale ($k_{ex} > \Delta\delta$, *Figure 2e–f*, *Prestel et al., 2018*). The change in peak position could be followed throughout the titration, and 2D NMR lineshape analysis resulted in affinity and dissociation rate constants of $K_d = 16 \pm 1$ µM and $k_{off} = 2600 \pm 100$ s$^{-1}$, respectively (*Figure 2—figure supplement 1b*, *Table 2*). Again, the CSPs spanned both CaM lobes (*Figure 2h*), although they were less pronounced for the C-lobe and generally smaller compared to those induced by H1 binding. Inverting the titration gave a similar result (*Figure 2—figure supplement 2b*). However, in this case the spectral quality of $^{15}$N-labeled H2 in both free and bound states was inferior with very heterogeneous signal intensities and only few assignable signals, likely originating from weak self-association (*Figure 2—figure supplement 3*). This will interfere with CaM interaction and provides a likely explanation for the affinity variations obtained from different titration setups.

Taken together, these data show that each of the CaM lobes can bind one NHE1 H1 with high affinity forming a 1:2 complex, with the C-lobe energetically slightly favored. CaM binds NHE1 H2 with lower affinity in a 1:1 complex with both CaM lobes involved, suggesting a more classical CaM binding as depicted in the cartoon in *Figure 2g*.

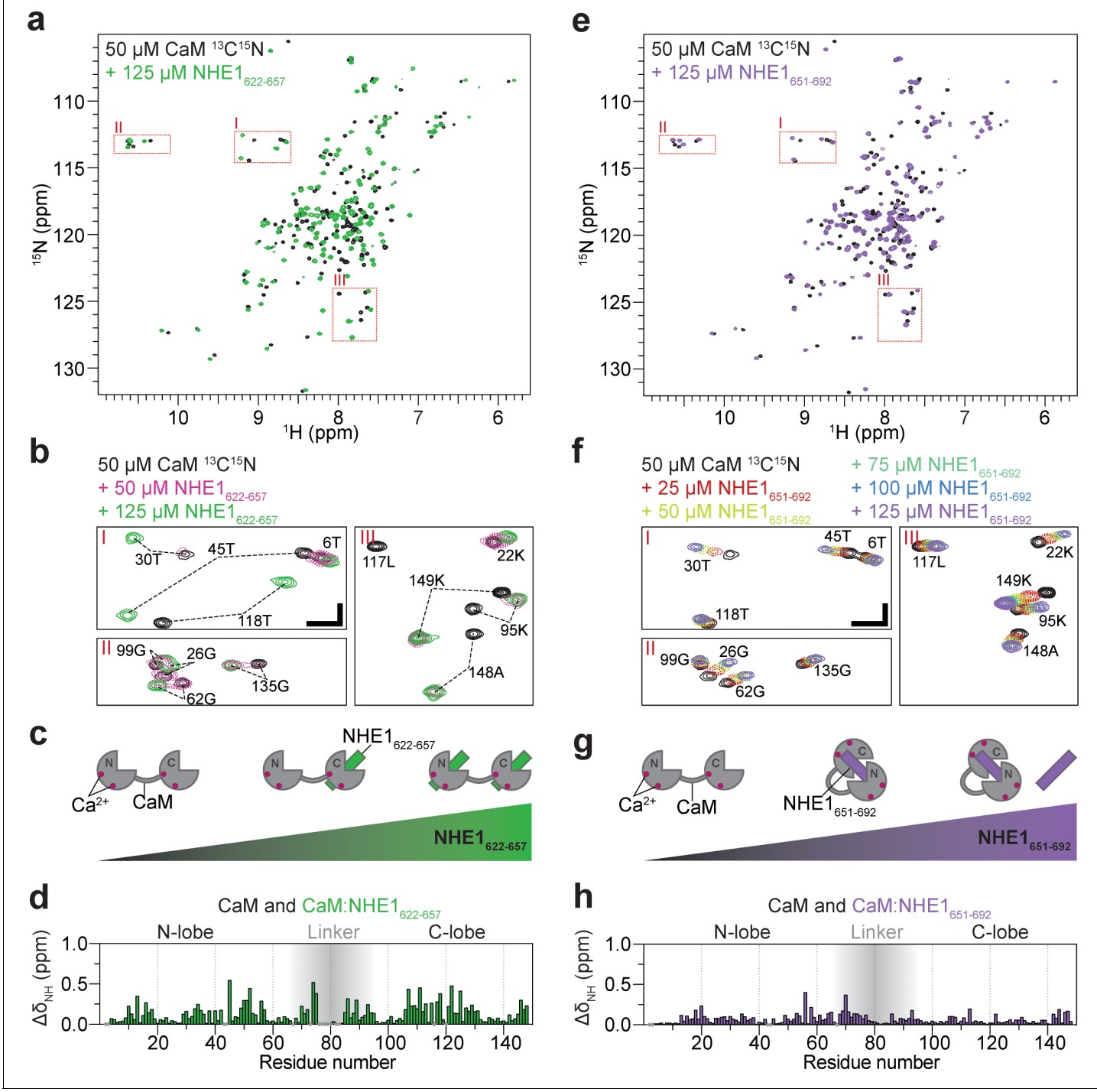

**Figure 2.** Calmodulin (CaM) interaction with NHE1$_{622-657}$ (H1) and NHE1$_{651-692}$ (H2). (**a**) $^1$H,$^{15}$N HSQC spectra of 50 µM $^{15}$N-CaM alone (black) and in the presence of 125 µM unlabeled NHE1$_{622-657}$ (H1, green). (**b**) Zoom on the three highlighted regions from the spectrum in (**a**); $^1$H,$^{15}$N HSQC spectra of 50 µM $^{15}$N-CaM in the presence of 0 µM (black), 50 µM (dashed red), and 125 µM (green) unlabeled NHE1$_{622-657}$. The horizontal and vertical scale bars correspond to 0.1 ppm δ$^1$H and 0.5 ppm δ$^{15}$N, respectively. (**c**) Cartoon representation of CaM interaction with increasing amounts of NHE1$_{622-657}$. (**d**) Chemical shift perturbations (CSP; ΔδNH) of CaM upon interaction with NHE1$_{622-657}$ at saturation (125 µM NHE1$_{622-657}$, 50 µM $^{15}$N-CaM, denoted 2:1 NHE1:CaM). (**e**) $^1$H,$^{15}$N HSQC spectra of 50 µM $^{15}$N-CaM alone (black) and in the presence (purple) of unlabeled NHE1$_{651-692}$. (**f**) Zoom on the three highlighted regions from the spectrum in (**e**); $^1$H,$^{15}$N HSQC spectra of $^{15}$N-CaM titrated from 0 to 125 µM (black to purple) with unlabeled NHE1$_{651-692}$ (H2). The horizontal and vertical scale bars correspond to 0.1 ppm δ$^1$H and 0.5 ppm δ$^{15}$N, respectively. (**g**) Cartoon representation of CaM interaction with increasing amounts of NHE1$_{651-692}$. (**h**) CSP (ΔδNH) of CaM upon interaction with NHE1$_{651-692}$ at saturation (125 µM NHE1$_{651-692}$, 50 µM $^{15}$N-CaM). HSQC - Heteronuclear Single Quantum Coherence; DMSO - Dimethylsulfoxide; RMSD - Root-mean-square deviation; NOE - Nuclear overhauser

*Figure 2 continued on next page*

*Figure 2 continued*

effect; BSA - Bovine serum albumine; SDS - Sodium dodecylsulfate; IPTG - Isopropyl-β-D-thiogalactoside; DSS -2,2-Dimethyl-2-silapentane-5-sulfonate sodium salt.

The online version of this article includes the following figure supplement(s) for figure 2:

**Figure supplement 1.** TITAN 2D nuclear magnetic resonance lineshape analysis for extraction of binding kinetics.

**Figure supplement 2.** Titration of $^{15}$N-labeled H1, H2, and H1H2 with unlabeled calmodulin (CaM).

**Figure supplement 3.** Concentration dependence of the nuclear magnetic resonance chemical shift of Na$^+$/H$^+$-exchanger (NHE1) H2.

## CaM can interact with NHE1 H1H2 in different conformations with different stoichiometries

We now revisit the complex between CaM and H1H2 by NMR. First, we titrated unlabeled H1H2 into $^{15}$N-labeled CaM and compared the $^{15}$N HSQC fingerprint spectra at different stoichiometries to those obtained with H1 or H2 alone. At a molar ratio of 1:2 (CaM:H1H2), the $^{15}$N HSQC spectrum closely resembled the state where only H1 is present in twofold excess (*Figure 3a–c*). This suggested that at this stoichiometry each CaM lobe interacts with one H1 of H1H2 in a ternary complex, while H2 is not involved in the interaction, as depicted in *Figure 3a*. At a stoichiometry of 1:1, the signals of the N-lobe of CaM still overlaid almost perfectly with the spectrum of $^{15}$N-labeled CaM at saturation with H1. However, signals of the C-lobe now overlaid with the spectrum of CaM bound to H2 (*Figure 3d–f*). Thus, at this molar ratio, the major state is the N-lobe of CaM bound to H1 and the C-lobe bound to H2, as depicted in *Figure 3d*. It is worth noting that a second set of very weak peaks was detectable for many of the residues. Comparing these to the spectra from the simpler systems of the individual peptides, this suggested the presence of a minor state with the opposite architecture: the N-lobe of CaM bound to H2 and the C-lobe bound to H1 (*Figure 3—figure supplement 1*). The severely broadened signals may thus be explained by exchange between these two states, and judging from the relative signal intensities, the equilibrium was highly shifted toward the H1:N-lobe;H2:C-lobe complex, as depicted in *Figure 3d*. A titration of $^{15}$N-labeled H1H2 with CaM supports these findings, although the spectral analysis was complicated by severe signal overlap (*Figure 2—figure supplement 2c, d*). At a stoichiometry of 1:2 (CaM:H1H2), two sets of signals originated from each residue of H1, while signals from H2 closely resembled those of unbound H2 (*Figure 2—figure supplement 2c*). This suggests a state with H1 bound to both CaM lobes, while H2 did not partake in the interaction. At a stoichiometry of 1:1, signals from H1 overlapped with the N-lobe bound state, while signals from H2 were either invisible or closely resembled the C-lobe bound state (*Figure 2—figure supplement 2d*). For both H1H2:CaM stoichiometries analyzed, the described complexes were the major states as judged by the relative NMR signal intensities. It is, however, important to note that at 1:1 as well as 1:2 (CaM: H1H2) stoichiometries the NMR signals were drastically broadened compared to those of free CaM or of the 1:2 complex saturated with H1 (*Figure 3a, d*), indicating a dynamic equilibrium of states with different stoichiometries, including potential higher order oligomers. Due to these challenges, it was not possible to determine the atomistic structure of the CaM:H1H2 complexes directly. Instead, to assess the absolute sizes of the complexes and support the proposed complex architectures (cartoons in *Figure 3a, d*), we used pulsed field gradient NMR diffusion measurements (PFG-NMR), dynamic light scattering (DLS), and size exclusion chromatography coupled to multiple angle light scattering (SEC-MALS). The results of these experiments are shown in *Figure 3g–i* and summarized in *Table 3*. The mass of the complex at a 1:1 stoichiometry as determined from SEC-MALS (25.0 ± 1.6 kDa) was in perfect agreement with the expected molecular weight of a 1:1 complex (25.5 kDa) and suggested that, if higher oligomers were present, their population was low under these conditions. Unfortunately, the

**Table 2.** Exchange kinetics from 2D NMR lineshape analysis.

| | $k_{on,1}$ (M$^{-1}$ s$^{-1}$) | $k_{off,1}$ (s$^{-1}$) | $K_{d,1}$ (nM) | $k_{on,2}$ (M$^{-1}$ s$^{-1}$) | $k_{off,2}$ (s$^{-1}$) | $K_{d,2}$ (nM) |
|---|---|---|---|---|---|---|
| H1 (two site) | $(1.5 \pm 0.6) \times 10^9$ | $40 \pm 5$ | 27* | $(3.7 \pm 1.3) \times 10^8$ | $16 \pm 3$ | 42* |
| H2 (one site) | $(1.6 \pm 0.1) \times 10^8$ | $2600 \pm 100$ | $16000 \pm 1000$ | – | – | – |

*$K_d$s taken from ITC analysis, kept constant during fitting.

NMR: nuclear magnetic resonance; ITC: isothermal titration calorimetry.

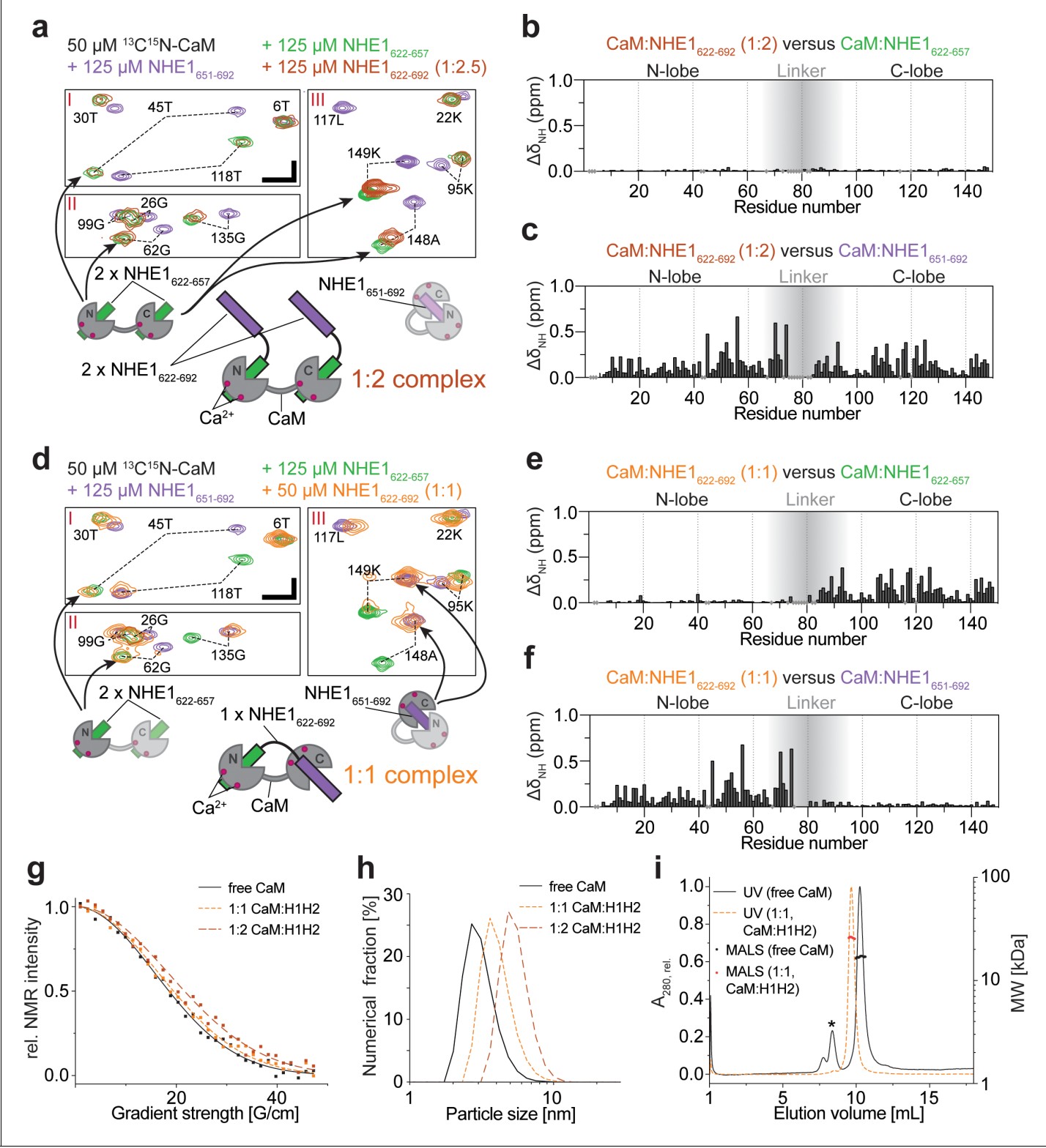

**Figure 3.** Calmodulin (CaM) interaction with NHE1$_{622-692}$ (H1H2) at two different stoichiometries. (**a**) The 1:2 CaM:NHE1$_{622-692}$ (H1H2) complex (molar ratio 1:2). Three regions of the $^1$H,$^{15}$N HSQC spectra of 50 µM $^{15}$N-CaM with 125 µM NHE1$_{622-657}$ (green), 125 µM NHE1$_{651-692}$ (purple), or 125 µM NHE1$_{622-692}$ (orange) are shown. The horizontal and vertical scale bars correspond to 0.1 ppm $\delta^1$H and 0.5 ppm $\delta^{15}$N, respectively. The complexes giving rise to the indicated peaks are depicted in the cartoons below (right and left), with the 1:2 complex in the center. (**b**) Chemical shift perturbations (CSPs) ($\Delta\delta$NH) between the 1:2 CaM:NHE1$_{622-692}$ complex and the 1:2 CaM:NHE1$_{622-657}$ complex (dark red and green spectra in **a**). The gray shade

*Figure 3 continued on next page*

Figure 3 continued

illustrates the linker region of CaM. (c) CSPs (ΔδNH) between 1:2 CaM:NHE1$_{622-692}$ and 1:1 CaM:NHE1$_{651-692}$ (dark red and purple spectra in a). (d) The 1:1 CaM:NHE1$_{622-692}$ complex. Three regions of the $^1$H,$^{15}$N HSQC spectra of 50 μM $^{15}$N-CaM in the presence of 125 μM NHE1$_{622-657}$ (green), 125 μM NHE1$_{652-692}$ (purple), or 50 μM NHE1$_{622-692}$ (orange). The horizontal and vertical scale bars correspond to 0.1 ppm δ$^1$H and 0.5 ppm δ$^{15}$N, respectively. Smaller cartoons represent the 1:2 CaM:NHE1$_{622-657}$ complex (left) and 1:1 CaM:NHE1$_{651-692}$ complex (right) and the 1:1 CaM:NHE1$_{622-692}$ complex identified from overlapping peaks in the $^1$H,$^{15}$N HSQC spectra. (e) CSPs (ΔδNH) between 1:1 CaM:NHE1$_{622-692}$ and 1:2 CaM:NHE1$_{622-657}$ (orange and green spectrum in d). (f) CSPs (ΔδNH) between 1:1 CaM:NHE1$_{622-692}$) and 1:1 CaM:NHE1$_{651-692}$ (orange and purple spectrum in d). The chemical shifts are available in BMRB accession code 34521. (g) Pulsed field gradient nuclear magnetic resonance diffusion measurements of 40 μM $^{15}$N-CaM without ligand (black), with 40 μM NHE1$_{622-692}$ (1:1, orange), and with 80 μM NHE1$_{622-692}$ (1:2, red). (h) Number-averaged particle size distribution from dynamic light scattering at a total protein concentration of 1 mg/mL in all three cases. (i) Size exclusion chromatography coupled to multiple angle light scattering elution profiles of free CaM (black, *bovine serum albumin from calibration run) and the 1:1 complex of CaM:H2 (orange).

The online version of this article includes the following source data and figure supplement(s) for figure 3:

**Source data 1.** Pulsed field gradient nuclear magnetic resonance raw data (relating to *Figure 3g*).
**Source data 2.** Raw dynamic light scattering correlation data (relating to *Figure 3h*).
**Source data 3.** Raw size exclusion chromatography coupled to multiple angle light scattering data (relating to *Figure 3i*).
**Figure supplement 1.** Minor population at a stoichiometry of 1:1 (CaM:H1H2).
**Figure supplement 2.** Influence of ethylenediaminetetraacetic acid (EDTA) on calmodulin (CaM) interaction with NHE1$_{622-692}$.

H1H2 peptide interacted with the column material when not fully saturated with CaM, which made the analysis of the free peptide as well as the 1:2 complex by SEC-MALS impossible. However, these complexes were readily accessible by DLS and PFG-NMR: both experiment series showed a stepwise increase of the complex size from free CaM to the complexes formed at 1:1 and 1:2 stoichiometries, respectively.

These results show that *in vitro* NHE1 and CaM can form multiple high-affinity complexes of different stoichiometries and structures, depending on the availability of each binding partner.

## Structure of the ternary complex of CaM and two NHE1 H1 helices

Our results suggested that CaM can act as an NHE1 dimerization switch at high [Ca$^{2+}$]$_i$, where each of the CaM lobes bind one H1, bridging two NHE1 tails. To obtain a more detailed understanding of this interaction, we determined the structure of the ternary complex of CaM bound to two H1 peptides using NMR, which is the major state in solution when either H1 or H1H2 are present in twofold excess (*Figure 3a, b*). Furthermore, the spectral quality of the complex with H1 was far superior to the exchange-broadened spectra of the CaM complex with H1H2. Therefore, the 1:2 complex of CaM and H1 constituted the foundation for obtaining the NMR constraints for structure determination. The results of the structure determination are shown in *Figure 4* and *Table 4*. Both CaM lobes have one H1 bound inside the hydrophobic cleft (*Figure 4a*). An overlay of the 10 lowest energy structures shows that the individual lobes are well defined (backbone RMSD: N-lobe + H1$_N$ = 0.77 ± 0.16 Å; backbone RMSD C-lobe + H1$_c$ = 0.83 ± 0.18 Å), while no NOEs between the two lobes could be obtained, suggesting that the linker region remains flexible in the bound state (*Figure 4b, c*). In both lobes, the interaction surface is defined by contacts between the hydrophobic face of the amphipathic H1 (L623, I631, I634, L635, N638, L639, T642, and L646; *Figure 4d, e*) and the central methionines in the hydrophobic cleft of each CaM-lobe. For the N-lobe, this involves M37, M52, M72, and M73; and for the C-lobe M110, M125, M145, and M146. In addition, a high complementarity in electrostatics is evident, with multiple positively charged residues on the opposite side of NHE1 H1 (K625, K637, R632, K633, R636, K641, R643, R645) complementing the negatively charged surface of CaM (*Figure 4f–i*). Although this

**Table 3.** Size evaluation of CaM-H1H2 complexes at different stoichiometries.

| | $D_{NMR}$ ($10^{10}$ m$^2$ s$^{-1}$) | $R_{h,NMR}$ (Å) | Number; volume; intensity averaged $R_{h, DLS}$ (Å) | Elution volume$_{SEC}$ (mL) | $MW_{MALS}$ (kDa) |
|---|---|---|---|---|---|
| Free CaM | 1.57 ± 0.05 | 20.8 ± 0.8 | 16 ± 1; 20 ± 1; 27 ± 2 | 10.27 ± 0.05 | 17.0 ± 0.9 |
| CaM:H1H2 (1:1) | 1.43 ± 0.04 | 22.8 ± 0.7 | 21 ± 2; 26 ± 2; 34 ± 2 | 9.68 ± 0.05 | 25.0 ± 1.6 |
| CaM:H1H2 (1:2) | 1.20 ± 0.05 | 27.2 ± 0.8 | 27 ± 2; 31 ± 2; 37 ± 2 | –* | –* |

*Not obtainable due to interaction of H1H2 with the SEC column material.
CaM: calmodulin; SEC: size exclusion chromatography; MALS: Multi-angle light scattering.

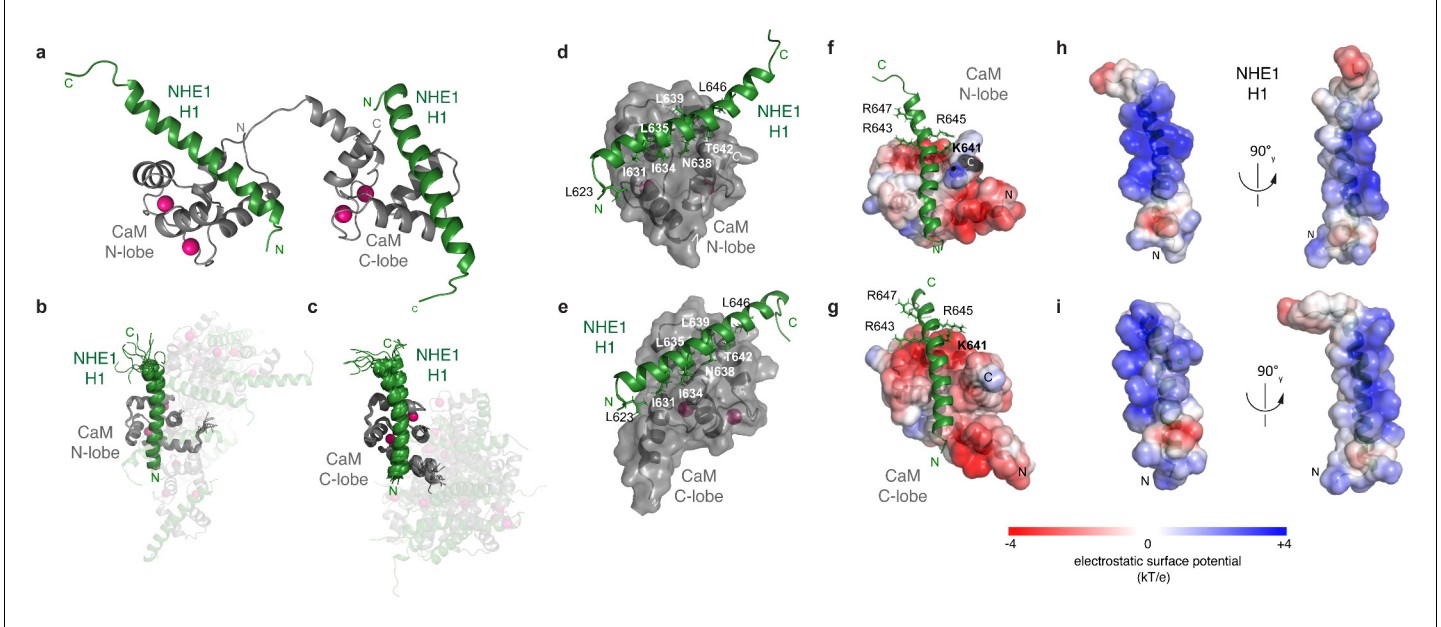

**Figure 4.** The ternary complex between calmodulin (CaM) and two Na+/H+-exchanger (NHE1) H1s. (**a**) Cartoon of the complex showing the two lobes of CaM in gray and the two NHE1 H1 helices in green. Ca2+ ions are shown in magenta. (**b, c**) Superimposition of the 10 lowest energy structures of the calculated ensemble aligned by (**b**) the CaM N-lobe with NHE1 H1 bound or (**c**) the CaM C-lobe with the second NHE1 H1 bound, showing mobility around the CaM linker. (**d, e**) Hydrophobic residues of NHE1 H1 forming contacts to CaM in the (**d**) N-lobe and (**e**) C-lobe, respectively. Electrostatic complementarity between Ca2+-CaM and NHE1-H1 in the (**f**) CaM N-lobe and the (**g**) CaM C-lobe shown by the surface potential of CaM and (**h, i**) for the two correspondingly bound H1 helices (PDB accession code 6zbi).

The online version of this article includes the following figure supplement(s) for figure 4:

**Figure supplement 1.** Comparison of the crystal structure (PDB: 2ygg) and the nuclear magnetic resonance (NMR) structure presented here (PDB: 6zbi).

complex is very different from that shown in the published crystal structure (*Köster et al., 2011*), inspection of the neighboring molecules in the crystal lattice reveals that the mode of interaction between the CaM C-lobe and H1, which we describe here by NMR, was also present in the crystal, but was translated as crystal contacts between symmetry-related molecules (*Figure 4—figure supplement 1*). The minor state observed by NMR at a stoichiometry of 1:1 (*Figure 3—figure supplement 1*) is similar to the crystal structure in that it has the same overall architecture (i.e., H1 bound to the hydrophobic cleft of the C-lobe and H2 bound to the hydrophobic cleft of the N-lobe). However, in the crystal structure, CaM binds to H1 and H2 from two different H1H2 molecules, while in solution we show by PFG-NMR, DLS, and SEC-MALS that the fraction of such higher oligomers is low at concentrations of 50 µM and below. The absolute protein concentration is certainly a parameter that can shift the equilibrium in this dynamic system, and the high concentrations used for crystallization might favor the formation of higher oligomers. Thus, collectively, these data depict a highly flexible system that is sensitive to the conditions under which they are studied.

## Influence of Ca2+ availability on complex formation

The experiments described so far were all performed under Ca2+ saturation. However, cellular [Ca2+]i is tightly regulated and numerous Ca2+-binding proteins compete for the available pool. To investigate the effect of limited Ca2+ availability on the structural ensemble of NHE1 and CaM, we monitored the state of CaM based on the 1H,15N HSQC fingerprint spectrum both under nominally Ca2+-free conditions and in the presence of the Ca2+ chelator ethylenediaminetetraacetic acid (EDTA). In the absence of Ca2+, we still observed CSPs of CaM after addition of 2.5-fold molar excess of H1H2 (*Figure 3—figure supplement 2a*), suggesting that also the Ca2+-free form of CaM binds NHE1 H1H2. Under conditions where both Ca2+ (here 2 mM) and high EDTA (here 6.6 mM) were present, free CaM gets stripped of Ca2+. However, in the additional presence of 2.5-fold molar excess of H1H2, signals of the C-lobe overlapped with those from Ca2+-CaM bound to H1, while signals of the N-lobe closely resembled the Ca2+-free state (*Figure 3—figure supplement 2b*). Thus,

**Table 4.** Structural statistics (PDB accession code 6zbi).

| Nuclear magnetic resonance restraints | |
|---|---|
| Distance, short-range $|i–j| \leq 1$ | 1019 |
| Distance, medium-range $1 < |i–j| < 5$ | 361 |
| Distance, long-range $|i–j| \geq 5$ | 224 |
| Distance, intermolecular | 213 |
| Dihedral restraints (Talos-N): | 336 |
| **Average pairwise RMSD** | |
| N-lobe + H1$_N$ <br> 8.73 (chain A), 623.644 (chain C) | Backbone: 0.77 ± 0.16 Å |
| | Heavy atoms: 1.42 ± 0.14 Å |
| C-lobe + H1$_C$ <br> 83.148 (chain A), 623.644 (chain B) | Backbone: 0.83 ± 0.18 Å |
| | Heavy atoms: 1.57 ± 0.17 Å |
| **Ramachandran statistics** | |
| Most favored regions | 95.7% |
| Additionally allowed regions | 3.7% |
| Generously allowed regions | 0.3% |
| Disallowed regions | 0.3% |

the presence of H1 increased the affinity for $Ca^{2+}$ in the C-lobe. This further suggests that competition for $Ca^{2+}$ favors another mode of interaction where the CaM C-lobe is bound to $Ca^{2+}$ and to H1 of NHE1, while the N-lobe is in its $Ca^{2+}$-free form.

These data show that $[Ca^{2+}]$ modulates the structural landscape of NHE1:CaM complexes further. Given the complexity of the cellular $Ca^{2+}$ landscape, especially at the plasma membrane–cytosolic interface where NHE1 functions, we therefore proceeded to address the functional role of NHE1: CaM interactions in a cellular context.

## In a cellular context, $Ca^{2+}$-dependent NHE1 activation is regulated by S648 phosphorylation status

In addition to the interaction with $Ca^{2+}$:CaM, cellular regulation of NHE1 is governed by the phosphorylation status of multiple sites in the C-terminal tail, some of which are located in the CaM-binding region (*Hendus-Altenburger et al., 2014*). S648, which can be phosphorylated by Akt (*Meima et al., 2009*; *Snabaitis et al., 2008*), is located at the C-terminal distal end of H1. In cardiomyocytes, the S648A mutation increased CaM binding to NHE1 *in vitro*, and S648 phosphorylation by Akt was proposed to inhibit NHE1 by preventing CaM binding (*Snabaitis et al., 2008*). In contrast, in fibroblasts, S648 phosphorylation by Akt was required for NHE1 activation by platelet-derived growth factor (*Meima et al., 2009*), indicating that the impact of this residue may be cell-type- and/or context-specific.

Reasoning that our new structural understanding of the NHE1:CaM complexes could help reveal the relation between $Ca^{2+}$:CaM, S648, and Akt in regulation of NHE1, we proceeded to investigate this issue. We first determined the effect of $Ca^{2+}$-CaM interaction and S648 phosphorylation on NHE1 localization and function. To this end, we generated stable clones of PS120 cells (a mammalian fibroblast cell line that lacks endogenous NHE1; *Pouyssegur et al., 1984*) expressing full-length WT hNHE1 and three variants. The 1K3R1D3E (K641D, R643E, R645E, R647E) variation introduces a charge reversal (CR) of four residues in H1, and will be referred to as H1-CR. A similar CR variant (1K3R4E) of hNHE1 was reported to exhibit reduced CaM-binding and increased $pH_i$ sensitivity, and fibroblasts expressing this variant showed reduced $Ca^{2+}$-induced NHE1 activation (*Bertrand et al., 1994*; *Wakabayashi et al., 1994*). Furthermore, MDA-MB-231 breast cancer cells expressing this CR variant (1K3R4E) showed increased invasion, migration, and spheroid growth compared to cells expressing WT NHE1 (*Amith et al., 2016*). The other two NHE1 variants expressed were S648A and S648D, representing the non-phosphorylated and phosphorylated states of S648, respectively. WT NHE1 and all three variants localized primarily to the plasma membrane – the expected pattern for

exogenously expressed NHE1 localization in PS120 cells (*Figure 5*; *Denker and Barber, 2002*; *Hendus-Altenburger et al., 2016*; *Hendus-Altenburger et al., 2019*). NHE1 protein expression levels were comparable between cell lines, albeit slightly higher in cells expressing the variants compared to WT, as shown by immunoblot analysis (*Figure 5b, c*). As expected, endogenous CaM exhibited similar expression levels in all cell lines (*Figure 5b, d*) and localized mainly diffusely in the cytosol (*Figure 5—figure supplement 1*).

Next, we determined the effect of CaM binding and S648 phosphorylation status on NHE1 ion transport activity. Cells were loaded with the pH-sensitive fluorescent probe BCECF-AM and subjected to real-time analysis of $pH_i$. The steady-state $pH_i$ was similar between all cell lines, ranging from ~7.4 to 7.5 (*Figure 5e*). To directly assess $Na^+/H^+$ exchange activity, we measured the rate of $pH_i$ recovery after an $NH_4Cl$-prepulse-induced acid load. Since PS120 cells lack endogenous NHE1 activity (*Figure 5f–i*, untransf.), $pH_i$ recovery in the nominal absence of $HCO_3^-$ reflects the activity of exogenously expressed NHE1 (*Pedersen and Counillon, 2019*). Following an acid load without further stimulation, recovery rates were not significantly different between cells expressing WT and variant NHE1s (*Figure 5f, g*). In the presence of the $Ca^{2+}$-ionophore ionomycin, which causes elevated $[Ca^{2+}]_i$ added concomitantly with the acid load, cells expressing WT NHE1 recovered significantly faster than WT vehicle controls (*Figure 5h, i*), in agreement with previous reports (*Wakabayashi et al., 1994*). Under ionomycin-stimulated conditions, recovery was significantly reduced in cells expressing the NHE1 H1-CR and S648D variants compared to those expressing WT NHE1 (*Figure 5h, i*).

These results show that mutations within the CaM-binding region (specifically H1) or mimicking S648 phosphorylation reduce $Ca^{2+}$-stimulated NHE1 activation in PS120 fibroblasts.

## Cellular NHE1:CaM proximity is resistant to H1-CR and S648 NHE1 mutations

To study the interaction between CaM and NHE1 *in situ,* we employed PLA, which detects close proximity (≤40 nm) of two proteins of interest *in situ* (*Söderberg et al., 2006*). Previous work has established the feasibility of detecting CaM interactions specifically using PLA and demonstrated that PLA signals between two binding partner proteins can be abrogated by mutation of one protein partner (*Edin et al., 2010*; *Ulke-Lemée et al., 2015*). *Figure 6a* shows the localization of NHE1:CaM PLA puncta (magenta) and NHE1 (green). Confirming assay specificity, PLA puncta were absent in native PS120 cells not expressing NHE1 (*Figure 6b, Figure 6—figure supplement 1*). In PS120 cells expressing WT NHE1, PLA puncta, representing ≤40 nM proximity of CaM and NHE1, were clearly detected (*Figure 6a, b*). The number of PLA puncta per area did not increase upon ionomycin treatment (*Figure 6a, b*), suggesting that an increase in $[Ca^{2+}]_i$ does not detectably increase the number of CaM and NHE1 proteins within close proximity of each other.

To determine the impact of S648 phosphorylation status on NHE1:CaM interaction in a cellular context, PS120 cells stably expressing WT NHE1, H1-CR, and the two phosphorylation variants, S648A and S648D, were also probed for NHE1:CaM proximity by PLA. In contrast to their inhibitory effect on $Ca^{2+}$-induced NHE1 activation (*Figure 5h, i*), none of the three variants abolished NHE1:CaM PLA *in situ* (*Figure 6a, b*). We therefore examined the solution NMR structure of the 1:2 (CaM:H1) complex and the positions of the different mutated residues. In the complex, the four residues K641, R643, R645, and R648 are all solvent exposed both when bound to the N-lobe or the C-lobe of CaM, with some hydrophobic contacts from K641 (*Figure 6c, d*). Likewise, in either of the lobes, S648 does not form direct contacts with CaM (*Figure 6c, d*). While several other mechanisms are possible (see Discussion), this may explain why the H1-CR, S648A, and S648D mutations have no apparent effect on NHE1:CaM proximity, although both H1-CR and S648D reduced $Ca^{2+}$-induced NHE1 activity.

These results show that despite their effects on $Ca^{2+}$-CaM-induced NHE1 activity neither H1-CR, S648A, nor S648D mutations abrogate the close proximity between NHE1 and CaM in a cellular context.

## $Ca^{2+}$-dependent NHE1 activation is dependent on CaM interaction but does not require Akt or PKC

In addition to its phosphorylation by Akt (*Meima et al., 2009*; *Snabaitis et al., 2008*), S648 is phosphorylated or predicted to be phosphorylated by several other kinases, including protein kinase C

(PKC) (*Hendus-Altenburger et al., 2014*). To investigate the phosphorylation of S648 by these kinases and study how it impacts NHE1:CaM interaction *in vitro*, we tried to phosphorylate the H1H2 peptide using either recombinant Akt or PKCδ in an *in vitro* phosphorylation assay. Phosphorylation of a single residue by Akt was confirmed by mass spectrometry, and S648 was identified by NMR, while PKCδ gave rise to mono-, double-, and triple phosphorylated forms of H1H2, identified by NMR at residues T642, S648, and T653 (*Figure 6h, Figure 6—figure supplement 2*). Thus, H1H2 is a potential target for other kinases. We used ITC to measure the affinity of monophosphorylated (pS648) H1H2 for CaM. As for the unphosphorylated H1H2, we observed two transitions in the binding isotherm and fitted this to a two-site binding model. The first binding event gave a $K_{d,1} = 5 \pm 4$ nM with a stoichiometry $n_1 = 0.93 \pm 0.02$, while the second binding event gave a $K_{d,1} = 2700 \pm 600$ nM with $n_2 = 1.16 \pm 0.06$ (*Figure 6i*, *Table 1*). Thus, indeed, phosphorylation of S648 results in a tenfold weakened binding to CaM compared to WT.

We next asked whether Akt or PKC were essential regulators of $Ca^{2+}$-dependent NHE1 activity. A 10 min stimulation with ionomycin had no detectable effect on Akt activation (phosphorylation) in PS120 cells (*Figure 6g*). Furthermore, Akt activity was ablated under both control and ionomycin conditions by the Akt inhibitor Akti-1/2 (*Figure 6g*), yet Akti-1/2 had no significant effect on ionomycin-stimulated NHE1 activity after an acid load (*Figure 6e*). Similarly, two well-established PKC inhibitors, bisindolylmaleimide I (BIM I) – targeting PKCα, -β1, -β2, -γ, -δ, and -ε – and Gö6983 – targeting PKCα, PKCβ, -γ, -δ, and -ζ (*Figure 6f*) – also had no significant effects on the $Ca^{2+}$-dependent NHE1 activity after acidification. These observations suggest that other kinases than Akt and these PKC isozymes can regulate S648 phosphorylation in the context of an increase in $[Ca^{2+}]_i$.

Our results show that S648 is phosphorylated by Akt and PKCδ *in vitro*, weakening H1H2 binding to CaM tenfold. However, under the conditions studied, inhibition of Akt or multiple PKC isoforms does not alter $Ca^{2+}$-induced cellular WT NHE1 activity. Thus, other kinases are likely important for S648 phosphorylation. Our findings also identify T642 and T653, present within the binding interface of the H1H2 NHE1 complex, as other phosphorylation sites potentially important for regulating NHE1:CaM interaction.

## Discussion

CaM is highly abundant in cells and its interactome comprises hundreds of proteins, the majority of which form 1:1 complexes with CaM (*Berchtold and Villalobo, 2014*; *Villalobo et al., 2018*). Most existing structures of CaM complexes have been determined by crystallography and X-ray diffraction, providing static representations of the interactions. Here, we used solution NMR and reveal that the energy landscape of the CaM complex with the $Na^+/H^+$ exchanger NHE1 – a key regulator of $pH_i$ in mammalian cells – is much broader than previously anticipated. In solution, it populates several different states with different affinities and structures, and the equilibrium between these states can be modulated by phosphorylation, NHE1:CaM ratio, and $[Ca^{2+}]_i$ (*Figure 7a, b*). *In vitro*, we found that NHE1 and CaM can form either 1:1 or 2:1 (NHE1:CaM) complexes with similar affinities but different structures, suggesting that the complex is highly dynamic and that the relevant state at any time is dependent on the availability of $Ca^{2+}$-CaM and NHE1 in the cell (*Figure 7*). In the absence of $Ca^{2+}$, apo-CaM can associate weakly with NHE1-binding sites in a different type of complex (*Figure 7c*). A recent study of the interaction between CaM and a truncated, minimal version of eEF-2K showed that the $Ca^{2+}$-free C-lobe of CaM bound the truncated eEF-2K in the absence of $Ca^{2+}$. Increasing $[Ca^{2+}]$ introduced an avidity effect through additional binding to the $Ca^{2+}$-loaded N-lobe (*Lee et al., 2019*). In a cellular context, where resting $[Ca^{2+}]_i$ is around 50–80 nM, a low $[Ca^{2+}$-CaM] scenario is more likely than complete absence of $Ca^{2+}$. Under such low $Ca^{2+}$ conditions, our results show that *in vitro* NHE1:CaM interaction can stabilize $Ca^{2+}$-binding to the C-lobe of CaM, resulting in an apo-state N-lobe and a $Ca^{2+}$-loaded C-lobe bound to H1 (*Figure 7d*). At higher $[Ca^{2+}$-CaM], corresponding to, for example, mitogenic stimuli, which can increase $[Ca^{2+}]_i$ to up to around 1 μM, the present study indicates that two scenarios are possible: one is a 1:1 NHE1:CaM complex, with the major state having the N-lobe bound to H1 and the C-lobe bound to H2 (*Figure 7e*), and the second is a 2:1 NHE1:CaM complex, with the CaM N- and C-lobes both interacting with H1 on two adjacent NHE1 C-tails, potentially stabilizing the dimeric state of NHE1 (*Figure 7f*). As CaM is limiting in a cellular context, the two latter scenarios will be modulated both by local $[Ca^{2+}]_i$ and CaM interactions with other proteins at any given time. Although the total

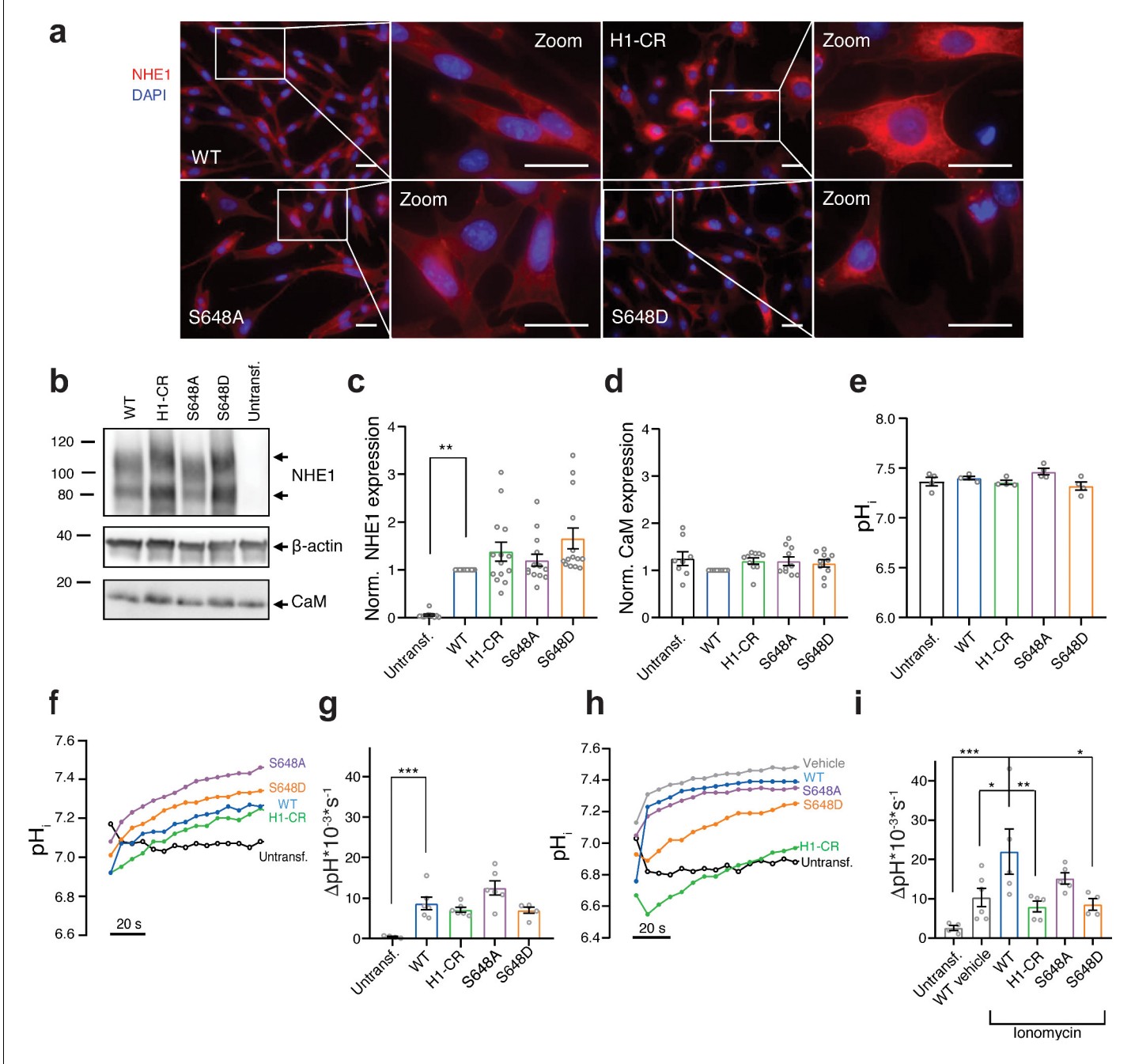

**Figure 5.** S648 phosphorylation status regulates $Ca^{2+}$-dependent $Na^+/H^+$-exchanger (NHE1) activation. (a) Representative immunofluorescence (IF) images of PS120 cells stably expressing wild-type (WT) or variant hNHE1 and stained for NHE1 (red) and DAPI (blue) to evaluate NHE1 expression and localization. Images were taken at ×60 magnification on an inverted Olympus Cell Vivo IX83 microscope, and areas marked with white squares are magnified. Scale bars represent 20 μm ($n = 5$). IF images of calmodulin (CaM) in all cell lines are shown in *Figure 5—figure supplement 1*. (b) Representative western blot of the four variant PS120 cell lines as well as untransfected PS120 cells blotted for NHE1 and endogenous CaM. For NHE1, the lower band corresponds to the immature protein and the upper band to fully glycosylated NHE1. β-actin was used as loading control ($n = 8–14$). (c, d) Total NHE1 ($n = 11–14$) or CaM ($n = 8–10$) expression was quantified using ImageJ and normalized for each variant to the loading control and then to the expression in the WT NHE1 cell line. (e) Steady-state $pH_i$ measured in Ringer solution in the absence of $HCO_3^-$ for each cell line. ($n = 4$). (f–i) Representative traces of BCECF-AM fluorescence converted to $pH_i$ for the four NHE1 variant cell lines and untransfected PS120 cells illustrating the time course of $pH_i$ recovery in Ringer in the absence of $HCO_3^-$ after an $NH_4Cl$-prepulse-induced acid load without (f, g) or with (h, i) the addition of 5 μM ionomycin or vehicle (EtOH). The recovery rates as $\Delta pH*10^{-3}*s^{-1}$ of each cell line are depicted in (g) ($n = 4–6$) and (i) ($n = 4–6$). The rate of $pH_i$ recovery is expressed as $\Delta pH/s$ because steady-state $pH_i$ and acidification were similar between cell lines, allowing direct comparison. It should be noted that the variant NHE1 proteins 1K3R1D3E, S648A, and S648D harbor a spontaneous F395Y mutation. The expression and $pH_i$ recovery rates of

*Figure 5 continued on next page*

Figure 5 continued

two WT NHE1s (two different clones) did not differ from that of F395Y (*Figure 5—figure supplement 2*). Data represent mean ± SEM. For (d, g, i), one-way analysis of variance with Dunnett's post-test comparing all cells to WT was carried out. For (c, e), Kruskal–Wallis non-parametric test comparing all cells to WT was carried out. Exact p-values stated for each group compared to WT were as follows: (c) untransf. p = 0.002, H1-CR p > 0.9999, S648A p > 0.9999, S648D p = 0.1054; (d) untransf. p = 0.8894, H1-CR p = 0.6035, S648A p = 0.5309, S648D p = 0.8852; (e) untransf. p > 0.9999, H1-CR p > 0.9999, S648A p > 0.9999, S648D p = 0.6911; (g) untransf. p = 0.0009, H1-CR p = 0.7781, S648A p = 0.1097, S648D p = 0.7652; (i) untransf. p = 0.0007, WT vehicle p = 0.0254, H1-CR p = 0.0089, S648A p = 0.3367, S648D p = 0.0195. *, ** and *** denote p < 0.05, p < 0.01 and p < 0.001, respectively.

The online version of this article includes the following source data and figure supplement(s) for figure 5:

**Source data 1.** Raw quantification data for Na$^+$/H$^+$-exchanger western blots (relating to *Figure 5b, c*).

**Source data 2.** Raw quantification data for calmodulin western blots (relating to *Figure 5b, d*).

**Source data 3.** Raw western blots (relating to *Figure 5b*).

**Source data 4.** pH$_i$ data summaries (relating to *Figure 5e–i*).

**Figure supplement 1.** Immunofluorescence staining of endogenous calmodulin (CaM) in PS120 cell lines stably expressing wildtype (WT) or variant hNHE1.

**Figure supplement 2.** Comparison of expression levels and pH$_i$ recovery rates of PS120 cells expressing two wildtype (WT) and F395Y hNHE1.

cellular [CaM] is 5–6 µM as measured in various cell types, the global free Ca$^{2+}$-CaM concentration in a cell has been estimated to be around 50 nM, that is, ~1% of the total [CaM] (*Persechini and Stemmer, 2002*; *Saucerman and Bers, 2012*). Both [CaM] and [Ca$^{2+}$]$_i$ may, however, reach much higher concentrations locally, for example, close to Ca$^{2+}$ channels (*Mori et al., 2004*), and the precise local Ca$^{2+}$-CaM availability around NHE1 will be dependent on local and global Ca$^{2+}$ signaling and competition with other CaM-binding proteins of various affinities. Our data suggest that in a cellular context CaM is bound to NHE1 at basal global [Ca$^{2+}$]$_i$. It is, however, possible that the local [Ca$^{2+}$]$_i$ immediately surrounding the NHE1 C-tail may reach substantially higher levels due to local complex formation with Na$^+$/Ca$^{2+}$ exchanger 1 (*Yi et al., 2009*). Hence, the precise Ca$^{2+}$ requirement for binding in a cellular context may be higher than the global [Ca$^{2+}$]$_i$.

CaM-mediated oligomerization has been reported for both soluble and membrane proteins. Examples of the former include the ERα, with two ERα binding to one Ca$^{2+}$-CaM, one in each lobe, to maximally activate transcription (*Li et al., 2017*; *Zhang et al., 2012*). The affinity of one CaM lobe for ERα is almost 100-fold lower than what we measured for H1 of NHE1, further supporting a unique role of the 1:2 CaM:NHE1 complex. For membrane proteins, both CaM-mediated multimerization and apo-CaM binding are well studied for tetrameric, voltage-gated K$^+$ channels. Small-conductance Ca$^{2+}$-activated K$^+$-(SK) channel monomers form 1:1 complexes with apo-CaM through C-lobe binding, thereby keeping the channel inactive. When [Ca$^{2+}$]$_i$ increases, the N-lobe also associates with the channel, triggering dimerization and channel opening (*Barros et al., 2019*; *Lee and MacKinnon, 2018*; *Schumacher et al., 2001*). Another example is the eag1/Kv10.1 channel, which is inhibited by CaM. Also for this interaction, the stoichiometry is 1:1, and it is proposed that CaM acts as a molecular clamp interacting with domains from neighboring subunits, thereby 'twisting' the pore closed (*Whicher and MacKinnon, 2016*).

NHE1 proteins are functional homodimers involving several incompletely understood mechanisms (*Pedersen and Counillon, 2019*). Based on our findings, it could be speculated that CaM binding contributes to NHE1 C-tail dimerization in a similar way as was observed for ERα, with CaM binding to H1 on two separate NHE1s in a 2:1 (NHE1:CaM) stoichiometry (*Figure 7f*), as shown in the ternary complex (*Figure 4*). Conceivably, the 1:1 stoichiometry observed for CaM:NHE1$_{622-692}$, with a C-lobe:H2 and N-lobe:H1 interaction, could also be possible through a 2:2 complex. This would require each CaM to span two NHE1 C-tails, binding to two different sites on two tails, forming a crossover structure (not shown). Whether this is feasible in a given cellular context will depend on the structural organization of the C-tail complexed with its binding partners and on the interaction of the NHE1 lipid interaction domain (LID) with the plasma membrane (*Hendus-Altenburger et al., 2020*). At the conditions studied here, we did not detect the 2:2 complex. However, a role for CaM as a dimerization switch would provide a structural understanding of previous reports of dimerization sites in the C-terminal tail of NHE1 (*Fafournoux et al., 1994*; *Hisamitsu et al., 2004*), as well as reports based on kinetic analyses of transport indicating that dimerization contributes to the regulation of NHE1 activity (*Fuster et al., 2008*; *Hisamitsu et al., 2006*; *Lacroix et al., 2004*; *Otsu et al., 1989*). Finally, in addition to the direct NHE1:CaM interactions studied here, the spectrum of NHE1:

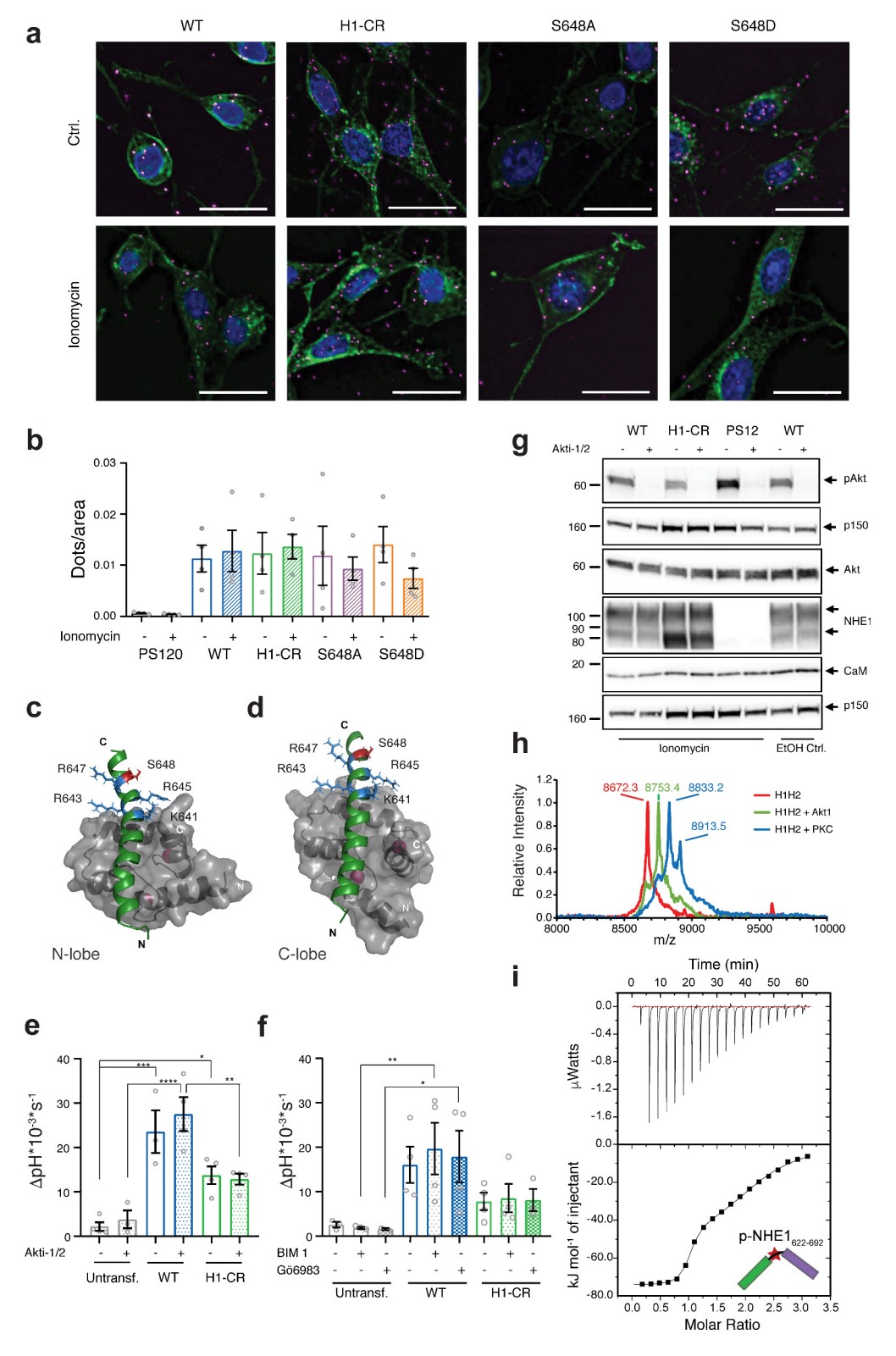

**Figure 6.** Cellular Na$^+$/H$^+$-exchanger:calmodulin (NHE1:CaM) proximity is resistant to H1-CR and S648 NHE1 mutations, and Akt and PKC activity does not significantly impact ionomycin-induced NHE1 activity. (a) Proximity ligation assay (PLA) of wildtype (WT) and variant hNHE1 in PS120 cells, with addition of vehicle (EtOH) or 5 µM ionomycin as shown ($n$ = 4–7). Negative controls are shown in *Figure 6—figure supplement 1* PLA signal is *Figure 6 continued on next page*

*Figure 6 continued*

magenta, co-staining for NHE1 is green, and nuclei (DAPI) are blue. Scale bars represent 20 µm. (**b**) Quantification of PLA signal in ImageJ after background subtraction and normalization to cell area. Approximately 400 cells were counted per condition per n (*n* = 4). Two-way analysis of variance (ANOVA) with Tukey's post-test comparing all conditions to one another showed no significant differences between groups. As there are 45 exact p-values for this figure, these are available upon request. (**c, d**) Structures of the individual lobes of the (2:1) NHE1 H1-CaM complex highlighting the position of the mutated residues and S648 in (**c**) the N-lobe and (**d**) the C-lobe. (**e**) Recovery rates as $\Delta pH * 10^{-3} * s^{-1}$ of each cell line with addition of 10 µM Akti-1/2 or vehicle (DMSO) (*n* = 4). Two-way ANOVA with Tukey's post-test was used to compare cell types to one another within each treatment group. Exact p-values were as follows: (**e**) Ctrl group: untransf. WT p = 0.0001, untransf. H1-CR p = 0.0141, WT H1-CR p = 0.0559; Akti-1/2 group: untransf. WT p < 0.0001, untransf. H1-CR p = 0.0792, WT H1-CR p = 0.0024. (**f**) As in (**e**), except that where indicated cells were treated with 5 µM of bisindolylmaleimide I (BIM) or 3 µM of Gö6983 (Go) (n = 4) or DMSO (vehicle). Exact p-values were as follows: Ctrl group: untransf. WT p = 0.0501, WT H1-CR p = 0.2291; BIM I group: untransf. WT p < 0.0091, WT H1-CR p = 0.0844; Go group: untransf. WT p < 0.0370, WT H1-CR p = 0.1823. (**g**) Western blot assessing the inhibition of Akt activity by Akti-1/2, assessed as phosphorylation of Ser473 of Akt (pAkt) compared to total Akt, in PS120 cells expressing WT and variant NHE1 (*n* = 4). The blot also illustrates endogenous levels of CaM and exogeneous NHE1. p150 was used as a loading control. NHE1 and CaM were detected on the same membrane as Akt. (**h**) Matrix-assisted laser desorption/ionization time of flight mass spectrometry analyses of NHE1-H1H2 before (red) and after addition of Akt (green) or protein kinase C (blue). (**i**) Representative binding experiments of Akt-phosphorylated NHE1-H1H2 titrated into CaM using isothermal titration calorimetry. The upper part shows baseline-corrected raw data from the titrations, and the lower part the normalized integrated-binding isotherms with the fitted binding curves assuming a two-site binding event. The peptide titrated into CaM is shown in cartoon, and the star indicates the phosphorylated S648. Data represent mean ± SEM. *, **, *** and **** denote p < 0.05, p < 0.01, p < 0.001, and p < 0.0001, respectively. The online version of this article includes the following source data and figure supplement(s) for figure 6:

**Source data 1.** Raw Na⁺/H⁺-exchanger calmodulin proximity ligation assays data (relating to *Figure 6a, b*).
**Source data 2.** pH$_i$ data summaries (relating to *Figure 6e, f*).
**Source data 3.** Raw western blots (relating to *Figure 6g*).
**Source data 4.** Isothermal titration calorimetry raw data and fits of triplicate experiments for the titration of calmodulin with pS648 H1H2 (*Figure 6h*).
**Figure supplement 1.** Lack of proximity ligation assays (PLA) signal in PS120 cells lacking Na⁺/H⁺-exchanger (NHE1).
**Figure supplement 2.** Identification of phosphorylated residues in Na⁺/H⁺-exchanger (NHE1) H1H2 by nuclear magnetic resonance.

CaM interactions *in vivo* additionally involves indirect interactions, such as within the complex between the disordered NHE1 C-tail and the CaM-binding Ser/Thr phosphatase calcineurin (*Hendus-Altenburger et al., 2019*; *Hisamitsu et al., 2012*).

Since the discovery that NHE1 is a CaM-binding protein (*Bertrand et al., 1994*; *Wakabayashi et al., 1994*), interaction with CaM has emerged as important for its regulation in response to a wide range of stimuli (*Coaxum et al., 2009*; *Li et al., 2013*; *Turner et al., 2007*). In most cases, however, the mechanisms involved have remained unaddressed. Initially, release of an autoinhibitory interaction with an allosteric 'pH sensor site' on NHE1 by CaM interaction was proposed (*Wakabayashi et al., 1994*). However, detailed kinetic analyses have questioned the existence of such a site and indicated that NHE1 activity is regulated by its dimerization state (*Fuster et al., 2008*; *Lacroix et al., 2004*; *Otsu et al., 1989*; *Pedersen and Counillon, 2019*). Our finding that CaM may contribute to stabilization of the NHE1 dimer state would reconcile these models. Finally, given the key role of anionic lipids including phosphatidyl-inositol(4,5)-diphosphate (PI[4,5]P$_2$) in regulating NHE1 by interacting with the C-tail (*Aharonovitz et al., 2000*; *Shimada-Shimizu et al., 2014*), Ca$^{2+}$-CaM could also regulate NHE1 through electrostatic tuning of the NHE1:PI(4,5)P$_2$ interaction. Such mechanisms are reported for several other membrane proteins (*Cao et al., 2013*; *Monteiro et al., 2014*) and would be consistent with the folding of the NHE1-LID domain upon membrane interaction, bringing the CaM-binding region in close proximity to the NHE1-LID and the membrane (*Hendus-Altenburger et al., 2020*).

The role of phosphorylation of S648 in NHE1 regulation has been a conundrum, with apparent conflicting findings in different cell types (*Meima et al., 2009*; *Snabaitis et al., 2008*). Here, we show that S648 phosphorylation, located at the distant end of H1, reduced NHE1:CaM interaction *in*

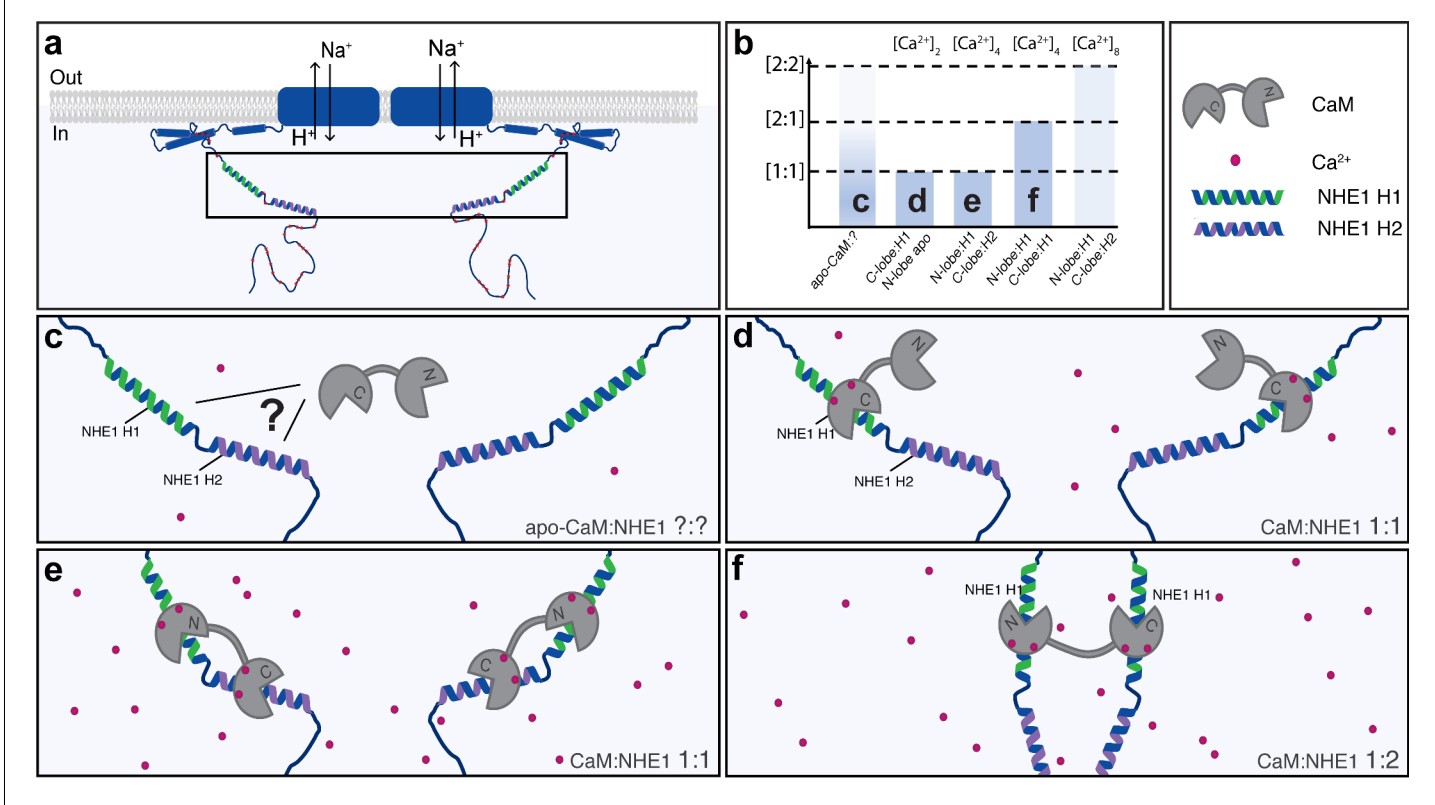

**Figure 7.** A dynamic view on Na$^+$/H$^+$-exchanger:calmodulin (NHE1:CaM) interaction. Multiple possible complexes exist depending on the [NHE1], [CaM], and [Ca$^{2+}$], including ternary complexes, suggesting that CaM may also contribute to NHE1 dimerization. (**a**) Schematic of NHE1 with a zoom on the CaM-binding region. The orientation of the monomers (N-terminal to N-terminal) in the NHE1 dimer is arbitrarily chosen. (**b**) Overview of the multiple complexes between CaM and NHE1 and their dependence on Ca$^{2+}$ (top). The indices (**c–f**) refer to the states shown in cartoon below. (**c**) Interaction between apo-CaM and NHE1 is seen, but its structure is unknown (indicated by a question mark). (**d**) At low Ca$^{2+}$ levels, NHE1-H1 is bound to the Ca$^{2+}$-loaded C-lobe of CaM, whereas the N-lobe is in its apo-state. (**e**) At high Ca$^{2+}$ levels, and high NHE1:Ca$^{2+}$-CaM ratio, a 1:1 complex is formed with H1 bound to the N-lobe and H2 bound to the C-lobe of CaM. (**f**) At high Ca$^{2+}$ levels, and low NHE1:Ca$^{2+}$-CaM ratio, a 1:2 complex is formed, where NHE1-H1 is bound to both the N-lobe and the C-lobe of CaM.

*vitro*, yet both the H1-CR variant and the two phosphorylation variants, S648A and S648D, retained close ($\leq$40 nm) proximity to CaM in cells. This may reflect that the effect of phosphorylation on the interaction was not strong enough to fully prevent interaction *in vivo*. Interestingly, the observation mirrors a recent study of CaM binding to smoothelin-like 1 (SMTNL1), where, CaM-SMTNL1 proximity was retained in cells upon mutation, despite loss of interaction *in vitro* (*Ulke-Lemée et al., 2015*). Our results agree with earlier work showing that cellular NHE1:CaM interaction is partially retained in an NHE1 variant functionally similar to our H1-CR variant (*Bertrand et al., 1994*). Additionally, cellular NHE1:CaM interactions may occur through C-tail interactions with other proteins (e.g., calcineurin) or membrane lipids (*Hendus-Altenburger et al., 2019*; *Hendus-Altenburger et al., 2020*), independent of the residues studied here but also giving rise to PLA signals. Finally, while NHE1:CaM proximity in the absence of interaction cannot be excluded, the observed binding of NHE1 to apo-CaM *in vitro* and the lack of effect of ionomycin on the number of proximity events observed in cells indicate that a fraction of NHE1:CaM complexes are present at all times.

In congruence with the reduced CaM affinity of the S648-phosphorylated H1H2 *in vitro*, Ca$^{2+}$-induced NHE1 activity in PS120 cells was reduced by the S648D phospho-mimicking variant. Inhibiting the kinase responsible for S648 phosphorylation would therefore be expected to increase Ca$^{2+}$-induced NHE1 activity. In our hands, an increase in [Ca$^{2+}$]$_i$ did not activate Akt, one obvious candidate kinase (*Meima et al., 2009*; *Snabaitis et al., 2008*), and inhibition of Akt had no significant effect on Ca$^{2+}$-mediated NHE1 activity. Furthermore, S648 was phosphorylated *in vitro* by both Akt and PKC, which have overlapping consensus sites, strongly indicating that other kinases than Akt can act via S648 to regulate NHE1. We reasoned that the conventional PKCs, which require an

increase in [Ca$^{2+}$]$_i$ for activation, were particularly relevant in this context. However, inhibition of conventional PKCs using two different inhibitors also had no effect on Ca$^{2+}$-induced NHE1 activation. Furthermore, as R643, R645, and R647, which are mutated in NHE1 H1-CR, also did not abrogate CaM:NHE1 complexes in cells and as they constitute key residues of the Akt and PKC consensus recognition site (*Rust and Thompson, 2011*), this can support an additional role for other kinases. Whether these phosphorylation sites are interdependent, potentially allowing a rheostatic response as, for instance, reported for the tumor suppressor p53 (*Lee et al., 2010*) and the cyclin-dependent kinase inhibitor Sic1 (*Borg et al., 2007*), remains to be addressed. T653 has not been studied in the context of CaM but has been implicated in NHE1 regulation by Rho kinase (*Wallert et al., 2015*) and β-Raf (*Li et al., 2016*). Thus, the mechanisms involved in physiological regulation of NHE1 via S648 (and T642 and T653) deserve further studies.

In conclusion, we show that NHE1 and CaM engage in diverse complexes of different NHE1:CaM stoichiometries and structures *in vitro*. The interactions can be dynamically tuned by variations in the concentrations of NHE1, CaM, and Ca$^{2+}$ and by phosphorylation of S648. In cells, Ca$^{2+}$-induced NHE1 activity was reduced by mutation in the first CaM-binding helix, as well as by mimicking S648 phosphorylation, but not by inhibition of Akt. In contrast, NHE1:CaM proximity was not lost by these mutations, likely reflecting further complexity of NHE1:CaM interactions *in vivo*. S648 phosphorylation *in vitro* was facilitated by both Akt and PKC, and Ca$^{2+}$-stimulated NHE1 activity was not affected by Akt inhibition, suggesting that multiple kinases regulate S648 phosphorylation *in vivo*. Finally, we determined the structure of the ternary complex between CaM and two NHE1 peptides, pointing to a possible role of CaM in stabilizing NHE1 dimers via binding to two C-terminal regulatory tails. Our results provide novel insight into the dynamics and regulation of NHE1:CaM interaction, and we suggest that an additional NHE1 regulatory layer tuned by Ca$^{2+}$-CaM availability may contribute to NHE1 dimerization. As CaM binding to membrane proteins is widespread, similar regulatory mechanisms and structural diversity may be relevant for tuning oligomerization and function in many other membrane proteins.

## Materials and methods

### Antibodies and reagents

Antibody against NHE1 (#sc-136239) was from Santa Cruz Biotechnology, Dallas, TX. Antibodies against CaM (#45689 and #05-173) were from Abcam, Cambridge, GB, and EMD Millipore, Burlington, MA, respectively. Antibodies against Akt (#9272) and pAkt (Ser473) (#4060) were from Cell Signaling Technology, Danvers, MA. Antibody against p150$^{Glued}$ (#610473) was from BD Transduction Laboratories, Franklin Lakes, NJ. Antibody against β-actin (#A5441) was from Sigma-Aldrich, St. Louis, MO. Anti-mouse (#P0447) and anti-rabbit (#P0448) HRP-conjugated secondary antibodies were from Dako, Glostrup, DK. Anti-mouse-Alexa Fluor 568 (#A10037), anti-rabbit- Alexa Fluor 488 (#A21206), and anti-mouse-Alexa Fluor 488 (#A21202) secondary antibodies for immunofluorescence were from Invitrogen, Carlsbad, CA. The Akt inhibitor Akti-1/2 (#124019) was from EMD Millipore, and the PKC inhibitors BIM I (cat# HY-13867) and Gö6983 (cat# HY-13689) were from MedChemExpress (Monmouth Junction, NJ).

### Cell lines and cell culture

PS120 cells (a kind gift from Laurent Counillon, University of Nice, France) were grown in DMEM (Life Technologies, Carlsbad, CA) supplemented with 5% (v/v) FBS (Sigma-Aldrich), 100 units, and 0.1 mg/mL Pen/Strep (Sigma-Aldrich) and 600 μg/mL G418 (EMD Millipore). The latter only for the transfected cells. Cells were grown at 37°C, 95% humidity, 5% CO$_2$, and passaged at a confluence of ~70–100%.

### Stable transfection of PS120 cells with variant NHE1

Transfections were performed with the Lipofectamine 3000 kit (Thermo Fisher, Waltham, MA). For each well in a 6-well plate, 125 μL medium with no supplements mixed with 3.75 μL Lipo3000 was added to a mixture of 125 μL medium with no supplements containing 4 μL P3000 reagent and 1 μg plasmid DNA. After 15 min incubation at room temperature (RT), 250 μL of the mixture was added to cells at 40–60% confluency without Pen/Strep and G418 and incubated 5 hr. After 5 hr, fresh medium containing Pen/Strep was added. The next day, fresh medium containing both Pen/Strep

and serum as described was added. At a confluence of ~75–95%, cells were split and grown in medium with the addition of 1900 μg/mL G418 (EMD Millipore). Medium was changed every 2–3 days and cells were split when needed until untransfected control cells had died. To ensure clonality, single colonies of cells arising from a single cell were picked using cell culture cylinders greased with Dow Corning high-vacuum silicon grease (Sigma-Aldrich). Briefly, cells were thinly seeded in 10 cm Petri dishes, washed in PBS, and cylinders placed on top of the colonies in question. Then, 50 μL 5% (v/v) trypsin (Sigma-Aldrich) was added, cells incubated in the incubator for 5 min, and 100–150 μL fresh medium added into the cylinders. Cell suspension from each cylinder was transferred to a well in a 6-well plate and medium added. After clonal selection, the concentration of G418 was decreased to 600 μg/mL.

## Immunofluorescence (IF) analysis

Cells were seeded on 12 mm coverslips in 6-well plates. At an ~70–90% confluence, cells were fixed in 100% MeOH for 15 min on ice and washed (3 × 5 min) in PBS on ice. Next, the coverslips were placed on parafilm and incubated with 0.5% (v/v) Triton X-100 in TBST (0.01 M Tris/HCl, 0.12 M NaCl, 0.1% Tween 20) for 15 min, blocked in 5% (w/v) BSA in TBST for 30 min, and incubated with primary antibodies diluted in 1% (w/v) BSA in TBST for 1.5 hr at RT. Coverslips were washed in TBST (3 × 5 min) and incubated with fluorophore-conjugated secondary antibodies in BSA/TBST for 1.5 hr. Coverslips were washed in TBST (5 × 5 min) including DAPI at dilution 1:1000 in BSA/TBST, mounted with N-propyl-gallate, sealed, and visualized using an inverted Olympus Cell Vivo IX83 with a Yokogawa CSU-W1 confocal scanning unit. Images were processed using ImageJ/Fiji.

## Immunoblotting

Cells were lysed (1% SDS, 10 mM Tris-HCl pH 7.5, 1 mM $Na_3VO_4$, cOmplete protease inhibitor cocktail (Sigma-Aldrich), heated to 95°C), sonicated for 2 × 30 s, and centrifuged for 5 min at 20,000 × $g$ at 4°C. Total protein amount was determined (DC Protein Assay kit, Bio-Rad, Hercules, CA) and samples equalized with $ddH_2O$ and NuPAGE LDS 4× Sample Buffer (Invitrogen) with dithiothreitol (DTT) added to final concentrations of 1× (sample buffer) and 125 mM DTT. Samples were run on precast Criterion TGX 10% gels (BioRad) in Tris/Glycine SDS buffer (BioRad) with Benchmark ladder (Invitrogen) and transferred to nitrocellulose membranes (BioRad). Membranes were Ponceau S (Sigma-Aldrich) stained, blocked (5% w/v nonfat dry milk in TBST), and incubated with primary antibodies overnight (o/n) at 4°C. Then, membranes were washed in TBST for 30 min, incubated with HRP-conjugated secondary antibodies for 1.5–2 hr, washed in TBST for 30 min, and developed with Pierce ECL Western Blotting Substrate (Thermo Fisher). Band intensities were quantified using ImageJ and normalized to the respective controls.

## Measurements of $pH_i$

Measurements of $pH_i$ were performed using the ammonium pre-pulse technique. Briefly, $8 * 10^4$– $12 * 10^4$ PS120 variant cells were seeded in 24-well plates 24 hr prior to the experiment, then loaded with 2′,7′-bis-(2-carboxyethyl)−5-(and-6)-carboxyfluorescein acetoxymethylester (BCECF-AM, 1.6 μM, Life Technologies) for 30 min at 37°C. Cells were washed in Ringer solution (in mM, 130 NaCl, 3 KCl, 20 HEPES, 1 $MgCl_2$, 0.5 $CaCl_2$, 10 NaOH pH 7.4) twice, then bathed in Ringer solution again and placed in a FluoStar Optima plate reader thermostated at 37°C. Emission was measured at 520 nm and excitation at 485 nm. After baseline measurement (10 min) in Ringer solution, acidification was induced by washout of either 20 mM $NH_4Cl$ after 5 min exposure (*Figure 5f*) or 5 mM after 25 min exposure (*Figure 5h*). $Na^+$-free solution (in mM, 135 NMDGCl, 3 KCl, 20 HEPES, 1 $MgCl_2$, 0.5 $CaCl_2$, pH 7.4) was applied (1.5 min), then the recovery was measured in Ringer solution for 10 min (*Figure 5f*). For the ionomycin experiment, 5 μM ionomycin (Sigma-Aldrich) or 96% EtOH (final dilution 0.48% v/v) was added to the $Na^+$-free and normal Ringer solutions during recovery and recovery was measured for 5 min (*Figure 5h*). For the Akti-1/2, BIM I, and Gö6983 experiments, 10 μM Akti-1/2 or DMSO (final dilution 0.11 % v/v) or 5 μM BIM I, 3 μM Gö6983 or DMSO (final dilution 0.2 % v/ v) as indicated was added to the cells while incubating for 30 min with BCECF-AM as well as to the Ringer and $NH_4Cl$ solutions and 5 μM ionomycin or 96% EtOH (final dilution 0.48% v/v) was added to the $Na^+$-free and Ringer solutions during recovery, which was measured for 5 min (*Figure 6e–f*). Calibration was performed using the high-$K^+$/nigericin technique (high $K^+$ Ringer: in mM, 140 KCl,

10 HEPES, 1 MgCl$_2$, 0.5 CaCl$_2$, 50 µM nigericin, pH 7.4) and an eight-point calibration curve obtained by fitting to the expression $\frac{F_{ave,pH=x}}{F_{ave,pH=7.4}} = 1 + b * (\frac{10^{pH-pK}}{1+10^{pH-pK}} - \frac{10^{7.4-pK}}{1+10^{7.4-pK}})$, where b and $pK$ are constants used for the conversion from fluorescence to pH $pH = pK - log_{10}(\frac{b}{y-1+b*a} - 1)$, where $\alpha = \frac{10^{7.4-pK}}{1+10^{7.4-pK}}$ (*Boyarsky et al., 1988*; *Pedersen et al., 1998*). Maximal acidification did not differ significantly between cell lines. Recovery rate was determined by fitting data to the expression IF(x<x$_0$, y$_0$) and calculating the derivative at 6 s after addition of Ringer recovery solution. Recovery rate for PS120 untransfected controls as well as a few traces that were not properly fitted using the expression above (H1-CR in n = 2 and n = 7 in *Figure 5i*; H1-CR BIM 5 µM in n = 3 and H1-CR in n = 4 in *Figure 6f*) was calculated as the slope of the initial approximately linear part of the curve.

## Proximity ligation assay

Cells were seeded on coverslips and treated for 10 min with either 96% EtOH (final dilution 0.48% v/v) in Ringer solution or 5 µM ionomycin in Ringer solution at a confluence of ~50–70%. Then, cells were fixed and permeabilized as described for IF analysis. The assay was carried out using Duolink PLA kit (Sigma-Aldrich) following manufacturer's protocol. Briefly, blocking solution was added for 25 min at RT, coverslips were then incubated with primary antibodies for 1 hr at 37°C, and washed 3 × 5 min in TBST in a 6-well plate with gentle shaking. PLA probes (PLUS and MINUS) were then added and incubated for 1 hr at 37°C, followed by washing as described. Ligase solution was added and incubated for 30 min at 37°C, followed by washing. Amplification solution containing polymerase was then added and incubated for 100 min at 37°C, followed by washing protected from light. In some experiments, green anti-mouse fluorophore-conjugated secondary antibody was applied o/n at 4°C to stain NHE1. The next day, coverslips were washed and DAPI diluted 1:1000 in ddH$_2$O was added for 5 min at RT, followed by washing. Coverslips were then mounted with N-propyl-gallate, sealed, and visualized using an inverted Olympus Cell Vivo IX83 with a Yokogawa CSU-W1 confocal scanning unit or, for z-stacks, an Olympus BX63. Coverslip identity was blinded for the observer. Confocal images were captured in a single plane across a grid of each coverslip. Images were quantified with ImageJ using the Analyze Particles function on binary images with background subtraction. Cells on coverslips used for quantification (*Figure 6b*) were not stained for NHE1. Approximately 400 cells were counted per condition per experiment. Cell sizes were measured manually using merged z-stacks of coverslips stained for NHE1 (*Figure 6a*).

## Protein expression and purification

A modified pET24a-vector coding for His-SUMO-tagged human NHE1 A622-D657 (H1), NHE1 R651-D692 (H2), or NHE1 A622-D692 (H1H2) was transformed into *Escherichia coli* BL21 Rosetta 2 (DE3) pLysS cells and grown o/n at 37°C in 10 mL LB media with 50 µg/mL kanamycin (kan) and 35 µg/mL chloramphenicol (cam). For expression of unlabeled NHE1 peptides, the cultures were added to 1 L LB media with 50 µg/mL kan and 35 µg/mL cam and grown at 37°C, 185 rpm to an OD$_{600}$ of 0.6–0.8, and induced with 0.5 mM IPTG and harvested after 3 hr by centrifugation at 4700 × *g* for 10 min at 4°C. For expression of stable isotope-labeled NHE1 peptides, the o/n culture was added to 1 L M9 minimal media (22 mM KH$_2$PO$_4$, 42 mM Na$_2$HPO$_4$·2H$_2$O, 17 mM NaCl, 1 mM MgSO$_4$), added 1:1000 of M2 trace element solution, 20 mM glucose (if required: [13]C-labeled), 19 mM NH$_4$Cl (if required: [15]N-labeled) with 50 µg/mL kan and 35 µg/mL cam, and grown similar to LB media cultures. After resuspension in 25 mL 50 mM Tris-HCl, pH 7.4, and centrifugation at 5000 × *g* for 15 min at 4°C, the pellet was dissolved in in 25 mL buffer NA (50 mM Tris-HCl, 10 mM imidazole, 150 mM NaCl, pH 8.0) supplemented with one tablet EDTA-free protease inhibitor (Roche). The cell solution on ice was sonicated 3 × 1 min at 90% amplitude (0.5 s cycles) using a UP200S Ultrasonic Processor (Hielscher), with 1 min breaks in between to cool down the sample. Subsequently, phenylmethylsulfonyl fluoride was added to the sample at a final concentration of 1 mM and centrifuged at 20,000 × *g* for 45 min at 4°C and the pellet discarded. A total of 5 mL Ni$^{2+}$ Sepharose 6 Fast Flow (GE Healthcare) was transferred to a 20 mL gravity flow column. The lysate supernatant was added to the column and incubated for 1 hr at RT under gentle shaking. The column was washed with 50 mL buffer NB (50 mM Tris-HCl, 10 mM imidazole, 1 M NaCl, pH 8.0), followed by a wash with 50 mL buffer NA. The protein was eluted with 10 mL buffer NC (50 mM Tris-HCl, 300 mM imidazole, 150 mM NaCl), and subsequently β-mercaptoethanol was added to a final concentration

of 3 mM together with ~100 µg ubiquitin-like-specific protease and incubated for 1 hr at RT. After cleavage, the NHE1 peptides were further purified by means of reversed-phase chromatography. A 17.35 mL Zorbax 300 Å StableBond C18 column was utilized using an Äkta Purifier 10 HPLC system. Trifluoroacetic acid (TFA) was added to the samples containing NHE1 peptides to a final concentration of 0.1% and centrifuged at 20,000 × g for 20 min at RT to remove insoluble aggregates prior to application. The column was equilibrated with 100% buffer RA (0.1% TFA [v/v] in MQ water) and a total volume of 5 mL was applied over two injections. A step gradient from 0% to 100% buffer RB (0.1% TFA [v/v], 70% acetonitrile [v/v] in MQ water) was set over 10 CV with a flow of 3.5 mL/min. $A_{280}$ was measured and fractions of 0.5–2.5 mL were collected and analyzed by SDS-PAGE. Fractions containing NHE1 peptides were pooled, flash-frozen, lyophilized o/n, and stored at −20°C.

A pET5-vector coding for full-length bovine CaM (1–149, 100% protein sequence identity to human CaM) was transformed into *E. coli* BL21 CodonPlus (DE3)-RP cells and grown o/n at 37°C in 10 mL LB media with 100 µg/mL ampicillin (amp) and 35 µg/mL cam. One liter of culture medium (LB or M9) was inoculated with 10 mL of the o/n culture, and expression was induced with 0.5 mM IPTG at an OD ~0.6. After induction, the cells were grown for 4 hr at 37°C and harvested by centrifugation at 6000 × g for 10 min. The cell pellet was suspended in 30 mL loading buffer (20 mM Tris-HCl, 500 mM NaCl, 10 mM $CaCl_2$, pH 7.5) supplemented with one pellet EDTA-free protease inhibitor (Roche), and sonicated on ice for 5 × 30 s at 80% amplitude (0.5 s cycles) using a UP200S Ultrasonic Processor (Hielscher), with 30 s breaks in between to cool down the sample. Subsequently, the lysate was centrifuged at 20,000 × g for 45 min at 4°C and the pellet discarded. The lysate supernatant was filtered using a 0.45 µm syringe filter and loaded onto a 5 mL phenyl Sepharose column. The column was subsequently washed with 40 mL loading buffer before eluting the protein in fractions of 2 mL using a 10 mL gradient from 0–100% elution buffer and additional 10 mL 100% elution buffer (20 mM Tris-HCl, 500 mM NaCl, 50 mM EDTA, pH 7.5). $A_{280}$ was measured and fractions were collected and analyzed by SDS-PAGE. The samples containing CaM were concentrated to ~1 mM, and the buffer was exchanged to the Gelfitration buffer (20 mM Tris, 100 mM KCl, optional 2 mM $CaCl_2$ depending on whether the $Ca^{2+}$ bound or unbound form was desired, pH 7.5) using an Amicon centrifugal filter (10 kDa cutoff) and applied to a HiLoad16/600 Superdex75 column. $A_{280}$ was measured and fractions were collected and analyzed by SDS-PAGE. Fractions containing CaM were pooled and stored at −20°C.

## NMR spectroscopy

All NMR spectra were recorded on Bruker Avance III 600 MHz or 750 MHz spectrometers equipped with TCI cryo-probes. If not specified otherwise, the sample conditions were 20 mM Tris, 100 mM KCl, 2 mM $CaCl_2$, 5% $D_2O$, 125 µM DSS (adjusted to pH 7.5 with 1 M HCl) at a temperature of 37°C. All spectra were referenced to DSS in the $^1H$ direct dimension and indirectly in the $^{15}N$ and $^{13}C$ dimensions using the gyromagnetic ratios. All spectra were zero-filled, apodized using a cosine bell window function in all dimensions, Fourier transformed, and phase corrected manually using either TopSpin3.6, NmrPipe (*Delaglio et al., 1995*), or qMDD (*Kazimierczuk and Orekhov, 2011*) if spectra were recorded using non-linear sampling. All spectra were analyzed and assigned manually using CCPNmr Analysis 2.4.2 (*Vranken et al., 2005*). For titrations experiments, series of $^1H,^{15}N$ HSQC NMR spectra of either protein ($^{15}N$-labeled) were recorded at 37°C in the absence and with increasing concentrations of the other up to 1:2.5 molar ratios, keeping the detected protein concentration constant.

## NMR assignment

Backbone and sidechain resonances of free CaM (sample A: 1.2 mM $^{13}C,^{15}N$-labeled CaM), free H1 (sample B: 0.5 mM $^{13}C,^{15}N$-labeled H1, recorded at 5°C), free H2 (sample C: 0.5 mM $^{13}C,^{15}N$ H2, recorded at 5°C), as well as of the CaM:H1 complex (sample D: 0.5 mM $^{13}C,^{15}N$-labeled CaM + 1.15 mM unlabeled H1; sample E: 1 mM $^{13}C,^{15}N$-labeled H1 + 0.5 mM unlabeled CaM) were assigned using sets of triple resonance spectra HNCO, HN(CA)CO, HNCACB, NH(CO)CACB, HcCH-TOCSY, hCCH-TOCSY, $^{15}N$-NOESY-HSQC, and $^{13}C$-NOESY-HSQC.

## 2D NMR lineshape analysis

The $^1H,^{15}N$ HSQC spectra were processed using an exponential weighting function in both dimensions (4 and 8 Hz line broadening in F1 and F2, respectively) and analyzed using the 2D lineshape

analysis tool TITAN for Matlab (*Waudby et al., 2016*) with a two-state binding model for the titration of CaM with H2. For the titration of CaM with H1, a model with two independent sites (four states) was used for fitting. To reduce the number of variables, the $K_d$s as determined from ITC were kept as constants in the fitting process and the assumption that the relaxation rates $R_2(^1H)$ and $R_2(^{15}N)$ of a residue in one lobe are not affected by peptide binding to the other lobe. In both cases, >20 isolated peaks were analyzed, and the errors were determined by a bootstrap analysis and are given by the standard deviation from the mean from 100 replicas.

## Structure calculations and refinements

Intra- and intermolecular distance restraints were obtained from samples D and E using a set of heteronuclear 3D NOESY experiments ($^{15}N$-NOESY-HSQC, $^{13}C$-NOESY-HSQC as well as $^{12}C/^{14}N$ filtered versions yielding exclusively intermolecular NOEs). The mixing time in all NOESY experiments was 120 ms. Backbone angular restraints were calculated from chemical shifts using Talos N (*Shen and Bax, 2013*). Initial NOE assignments and structure calculations for the CaM:H1 complex were done iteratively using Cyana 3.98.5 (*Güntert and Buchner, 2015*). Four $Ca^{2+}$ ions were included and the geometry of calcium coordination was fixed based on distance and angular restraints from crystal structures (PDB accession code: 1CLL; *Chattopadhyaya et al., 1992*). The structures were further refined with an implicit water model (eefxPot; *Tian et al., 2014*) using XPLOR-NIH v2.44 (*Bermejo and Schwieters, 2018*).

## Pulsed field gradient NMR diffusion experiments

For the determination of the diffusion coefficients by PFG-NMR, three samples of 40 μM of $^{15}N$-labeled CaM were prepared either without any ligand, with 40 μM H1H2 (1:1 complex), or with 80 μM H1H2 (1:2 complex), respectively. A series of $^{15}N$ filtered 1D $^1H$ spectra with preceding PFG-LED diffusion filter (*Wu et al., 1995*) were recorded on a Bruker Avance III 600MHz spectrometer equipped with a TCI cryo-probe, and the gradient strength was increased linearly from 2.4 to 47.2 G/cm. To determine the diffusion coefficients D, the decay curves of the integrated amide signals in the range of 7–9 ppm were plotted against the gradient strength and fitted in Dynamics Center (Bruker) using the equation:

$$I=I_0*exp(-D*x^2*\gamma^2*\delta^2(\Delta-\delta/3)*10^4)$$

where $I$ is the intensity of the NMR signal at the respective gradient strength, $I_0$ is the intensity without applied gradient, $x$ is the gradient strength in G/cm, $\gamma$ = 26752 rad/(G*s), $\delta$ = 3 ms, and $\Delta$ = 200 ms. $R_h$ was calculated from the diffusion coefficient using the Stokes–Einstein relation $R_h = k_BT/(6\pi\eta D)$ with $\eta$ = 0.6959 mPa * s (dynamic viscosity of water at 37˚C).

## Isothermal titration calorimetry

All ITC data were recorded on a Microcal instrument ITC200 (Malvern). Data were recorded at 25˚C with a stirring speed of 307 rpm and a reference power of 10 μCal/s. The buffer composition was 20 mM Tris (adjusted to pH 7.5 with 1 M HCl), 100 mM KCl, 2 mM $CaCl_2$, and all samples were degassed freshly before usage. For the high-affinity interactions (H1, H1H2, and H1H2 pS648), CaM was at an initial concentration of 10 μM in the sample cell and the peptides were added from a stock concentration of 150 μM in the syringe. The titration of H2 into CaM was conducted at 50 μM CaM in the cell and 670 μM H2 in the syringe. In this case, the heat of dilution was determined from a control experiment, where H2 was injected into a buffer solution, and this data set was subsequently used for baseline correction. Data from the ITC experiments were analyzed using the Origin 7 software package (MicroCalTM). The data sets were fitted to models for a single binding site (H1, H2) or two independent binding sites (H1, H1H2, H1H2 pS648). The extracted thermodynamic parameters (*Table 1*) are the mean and standard deviation of three independent experiments, and one representative experiment is shown in *Figure 1c–e*.

## Dynamic light scattering

DLS experiments were carried out on a Malvern Zetasizer NANO ZSP at a total protein concentration of 1 mg/mL. The data was analyzed with the Malvern Zetasizer software v8.1 using standard settings. The calculated values are given in *Table 3* as the average and standard deviation of three measurements.

## Size exclusion chromatography coupled to multiple angle light scattering

SEC-MALS was performed using an ÄKTA Pure (GE) system and a Superdex S75 10/300 column in line with a multiangle light scattering detector (miniDAWN TREOS, Wyatt Technologies) and a refractometer (Optilab T-rex, Wyatt Technologies). Data analysis was performed with the analysis software ASTRA (version 7.3.1.9 provided by Wyatt Technologies). An aliquot of 100 µL of either 50 µM CaM or 50 µM CaM+H1H2 (1:1 complex) was injected over the column at a flow rate of 0.5 mL/min in 10 mM Tris, 100 mM KCl, 2 mM $CaCl_2$, pH 7.4. The normalization, alignment, and band broadening were performed in ASTRA based on a BSA standard.

## Helix propensity

Helix propensity was predicted using Agadir (http://agadir.crg.es/) using standard settings (*Muñoz and Serrano, 1997*).

## *In vitro* phosphorylation and mass spectrometry

*In vitro* phosphorylation assays were carried out with 100 µM unlabeled or $^{13}C,^{15}N$-labeled H1H2 in 50 or 200 µL phosphorylation buffer (25 mM 3-(N-morpholino)propanesulfonic acid, 25 mM $MgCl_2$, 5 mM ethylene glycol-bis(β-aminoethyl ether)-N,N,N′,N′-tetraacetic acid, 2 mM EDTA, 10 mM adenosine triphosphate, 5 mM DTT, pH = 7.2), respectively. To this, 0.3 µg of active, GST-tagged, human Akt1 (Sigma-Aldrich) or 0.3 µg active, GST-tagged, human PKC δ (Biaffin GmbH) was added and incubated at 37°C for up to 120 hr. The unlabeled samples were analyzed by matrix-assisted laser desorption/ionization time of flight mass spectrometry on a Bruker autoflex III smartbeam spectrometer with α-cyano-4-hydroxycinnamic acid as matrix after 24 hr. The $^{13}C,^{15}N$-labeled samples were analyzed by NMR spectroscopy after 120 hr and partially assigned as described above.

## Statistical analysis

Data are shown as representative images or as means with standard error of means error bars as indicated. Statistical analysis was carried out with software GraphPad Prism 8. Statistical significance for experiments with one variable (e.g., cell type/group) was assessed using one-way analysis of variance (ANOVA) followed by Dunnett's or Tukey's post-test where appropriate (applied for $pH_i$ and protein expression data in *Figure 5*) or Kruskal–Wallis where appropriate (applied for steady-state $pH_i$ and protein expression in *Figure 5*). Statistical significance for experiments with two variables (e.g., cell type and treatment) was assessed using unmatched two-way ANOVA followed by Tukey's post-test (applied for $pH_i$ and PLA data in *Figure 6*). The comparisons carried out and the exact p-values from each test are indicated in the figure legends except for *Figure 6b*, where the 45 exact p-values are available upon request.

*, **, *** and **** denote $p < 0.05$, 0.01, 0.001, and 0.0001, respectively, unless otherwise specified.

# Acknowledgements

The authors thank Ruth Hendus-Altenburger for valuable discussion in the early phase of the study and Johs Dannesboe, Katrine Franklin Mark, Rikke Larsen Agersnap, and Signe A Sjørup for skilled technical assistance.

# Additional information

## Funding

| Funder | Grant reference number | Author |
|---|---|---|
| Danish Research Academy | 4181-00344 | Birthe B Kragelund |
| Novo Nordisk Fonden | NNF15OC0016670 | Birthe B Kragelund |
| Novo Nordisk Fonden | NNF18OC0034070 | Stine Falsig Pedersen |
| Novo Nordisk Fonden | NNF19OC0057241 | Stine Falsig Pedersen |

| Novo Nordisk Fonden | NNF18OC0032996 | Birthe B Kragelund |
| Villum Fonden | | Birthe B Kragelund |
| Carlsbergfondet | CF20-0491 | Stine Falsig Pedersen |
| Novo Nordisk | | Lise M Sjøgaard-Frich |

The funders had no role in study design, data collection and interpretation, or the decision to submit the work for publication.

## Author contributions

Lise M Sjøgaard-Frich, Conceptualization, Formal analysis, Validation, Investigation, Visualization, Methodology, Writing - original draft, Writing - review and editing; Andreas Prestel, Conceptualization, Formal analysis, Supervision, Validation, Investigation, Visualization, Methodology, Writing - original draft, Writing - review and editing; Emilie S Pedersen, Marc Severin, Formal analysis, Validation, Investigation, Visualization, Methodology, Writing - original draft, Writing - review and editing; Kristian Kølby Kristensen, Formal analysis, Investigation, Methodology, Writing - review and editing; Johan G Olsen, Formal analysis, Investigation, Writing - review and editing; Birthe B Kragelund, Conceptualization, Resources, Formal analysis, Supervision, Funding acquisition, Investigation, Visualization, Methodology, Writing - original draft, Project administration, Writing - review and editing; Stine Falsig Pedersen, Conceptualization, Resources, Formal analysis, Supervision, Funding acquisition, Visualization, Methodology, Writing - original draft, Project administration, Writing - review and editing

## Author ORCIDs
Andreas Prestel ⓘ https://orcid.org/0000-0002-5459-9608
Birthe B Kragelund ⓘ https://orcid.org/0000-0002-7454-1761
Stine Falsig Pedersen ⓘ https://orcid.org/0000-0002-3044-7714

## Decision letter and Author response
Decision letter https://doi.org/10.7554/eLife.60889.sa1
Author response https://doi.org/10.7554/eLife.60889.sa2

# Additional files

## Supplementary files
• Transparent reporting form

## Data availability
Source data files are provided for Figure 1, 3, 5 and 6. Resonance assignments of the ternary complex of CaM and two H1 have been deposited in the Biological Magnetic Resonance Bank (BMRB) under ID code 34521. The atomic coordinates for the ternary complex of CaM and two H1 have been deposited in the Protein Data Bank under the ID code 6zbi.

The following datasets were generated:

| Author(s) | Year | Dataset title | Dataset URL | Database and Identifier |
|---|---|---|---|---|
| Prestel A, Sjøgaard-Frich LM, Pedersen SF, Kragelund BB | 2021 | 1H, 13C and 15N chemical shift assignments of the 2:1 complex between NHE1 and calmodulin | http://www.bmrb.wisc.edu/data_library/summary/index.php?bmrbId=34521 | Biological Magnetic Resonance Data Bank, 34521 |
| Prestel A, Sjøgaard-Frich LM, Pedersen SF, Kragelund BB | 2021 | Ternary complex of Calmodulin bound to 2 molecules of NHE1 | http://www.rcsb.org/structure/6zbi | RCSB Protein Data Bank, 6ZBI |

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
