## [Decision Letter]

**Acceptance summary:**

The reviewers recognize the importance of structural dynamics studies of complexes, emphasizing that biological systems are not fixed, but rather interact dynamically with their targets. Your work focusing on the engagement of calmodulin with the Na/H exchanger NHE1, using a number of different biophysical methods, in particular NMR, provides a nice illustration of this, with multiple states modulated by the stoichiometries of the components. It further adds to the many different important roles of calmodulin in its interactions with targets.

**Decision letter after peer review:**

[Editors’ note: the authors submitted for reconsideration following the decision after peer review. What follows is the decision letter after the first round of review.]

Thank you for submitting your work entitled "Dynamic NHE1-Calmodulin complexes of varying stoichiometry and structure regulate Ca^2+^-dependent NHE1 activation" for consideration by *eLife*. Your article has been reviewed by a Senior Editor, a Reviewing Editor, and two reviewers. The following individual involved in review of your submission has agreed to reveal their identity: Walter J Chazin (Reviewer #2).

Our decision has been reached after consultation between the reviewers. Based on these discussions and the individual reviews below, we regret to inform you that your work will not be considered further for publication in *eLife*.

All of us are convinced of the importance of solution NMR in studies of the sort that you do involving multiple interactions and dynamic systems. Yet there were several flaws identified, including the lack of a structure of CaM bound to the full C-terminal cytoplasmic domain (or a fragment containing H1 and H2) and establishment of binding stoichiometry to full length NHE1. Is it a 1:2 or 2:2 complex, or a mixture of both? A 2:2 complex would agree with the crystal structure, whereas 1:2 stoichiometry would support the NMR structure of CaM/H1. Lastly, it was felt that the section on phosphorylation did not contribute substantially to the manuscript.

Reviewer #1:

The manuscript presents NMR, ITC and functional data for the protein calmodulin (CaM) and its interaction with NHE1. Importantly, the study suggests that CaM may bind to NHE1 in multiple ways and that CaM binding may stabilize dimerization of NHE1. The data and approach all seem satisfactory. However, I have a few serious concerns below about the interpretation of the NMR structure in Figure 4 (and implications in Figure 7), and the narrow scope of the approach, which all need to be addressed before a final evaluation can be made.

1) The NMR structure of CaM bound to the NHE1 H1 helix (residues 620-650) in Figure 4 provides a very narrow view of the CaM interaction with NHE1, and is difficult to interpret the wider significance of this structure without knowing the overall stoichiometry of CaM binding to the full-length NHE1. Is the overall stoichiometry of CaM bound to full-length NHE1 known? If not, then it needs to be determined in this study in order to properly interpret the structure in Figure 4. For example, does the full-length NHE1 dimer bind to two CaM (2:2 complex) or just one CaM (1:2 complex)? If the CaM/NHE1 stoichiometry is 2:2, then the crystal structure of CaM bound to H1H2 complex (2YGG) already explains how CaM could stabilize dimerization of NHE1. The crystal structure (Figure 4—figure-supplement 1) shows that the CaM C-lobe binds to H1 of one molecule from NHE1 dimer, while the CaM N-lobe binds to H2 of the second molecule from NHE1 dimer. In other words, the two CaM lobes are domain swapped in the NHE1 dimer, which reveals how CaM stabilizes NHE1 dimerization as a 2:2 complex. So, if the overall CaM/NHE1 stoichiometry is measured to be 2:2, then the crystal structure already nicely explains how CaM stabilizes dimerization of NHE1 and the NMR structure in Figure 4 would merely confirm what was already known from the crystal structure. Alternatively, if the overall stoichiometry turns out to be 1 CaM per NHE1 dimer (1:2 complex), then the prediction from the crystal structure about domain swapping may be wrong, and the NMR structure in Figure 4 will in that case become far more interesting. For example, if the overall stoichiometry is 1:2, then the structure depicted in Figure 7f would explain how CaM stabilizes dimerization by having each CaM lobe bind to H1 (while H2 does not bind to CaM). This is at complete odds with the 2:2 structure (crystal structure) in which the CaM C-lobe binds to H1 and CaM N-lobe binds to H2. A simple way to distinguish the 1:2 model (Figure 7F) from the 2:2 model (crystal structure) would be to delete H2 from the full-length NHE1. The deletion of H2 should not affect CaM-induced dimerization of NHE1 in the 1:2 model, because CaM does not bind to H2 in this model. However, the deletion of H2 should abolish CaM-induced dimerization of NHE1 in the 2:2 model. So, in addition to measuring the overall CaM/NHE1 stoichiometry, it will also be necessary to prepare the H2 deletion construct of NHE1 and see if this deletion mutant impairs CaM-induced NHE1 dimerization.

2) Why was the NMR structure only reported for CaM/H1 complex? It would be far better to report the NMR structure of CaM bound to H1H2 or the entire C-terminal cytosolic domain. Does CaM bind to H1H2 (or cytosolic domain) as a 2:2 complex in solution? In other words, does CaM induce dimerization of H1H2 in solution (like what is implied in the crystal structure)? Does the C-terminal cytosolic domain form a dimer in solution, and does CaM binding stabilize this dimerization?

3) Figure 3D shows a model of CaM bound to H1H2 in a 1:1 complex. This model predicts the CaM N-lobe binds to H1 and CaM C-lobe binds to H2, which is the exact opposite of what was shown in the crystal structure (CaM N-lobe binds to H2). The discrepancy here needs more explanation. How do you know that the CaM N-lobe is bound to H1 in the 1:1 complex here? The NMR structure of CaM bound to H1H2 is needed here to prove whether the N-lobe actually binds to H1 or H2 in the CaM/H1H2 complex.

4) Figure 7 shows various types of CaM/NHE1 interaction. Unfortunately, the most interesting interaction (2:2 complex predicted from the crystal structure) is not shown. An extra panel should be added in Figure 7 to show the 2:2 interaction (implied from the crystal structure), in which the CaM N-lobe binds to H2 from one molecule of NHE1 dimer, and the CaM C-lobe binds to H1 from the other molecule of NHE1 dimer. The 2:2 interaction should depict two CaM molecules interacting in a crisscrossed fashion with the NHE1 dimer. Again, it will be important here to experimentally test the 1:2 model (Figure 7F) versus the 2:2 model implied by the crystal structure. The H2 deletion described above in point 1 can distinguish these models. The most direct way to determine which model is functionally most relevant would be to measure the overall stoichiometry of CaM/NHE1. Is it 1:2 or 2:2 or perhaps a composite of the two? Without knowing the CaM/NHE1 stoichiometry, the functional relevance of the NMR structure in Figure 4 cannot be properly understood.

Reviewer #2:

This manuscript reports several interesting observations about the interaction of calmodulin with NHE1, a Na+/H^+^ transport channel. The experiments are well-designed and technically sound. However, these observations are not well connected to each other. What is the main point? Is it defining the physical interaction of calmodulin with NHE1, the mechanism of NHE1 S648 phophorylation, NHE1 activation, or something else? More importantly, it is not evident that a significant advancement has been made in understanding either the biochemical mechanism or biological function of system.

1) There is an obvious gap in logic that is not addressed, i.e. why the solution NMR structure was determined for only half of the NHE1 interaction domain. The ITC data clearly point to the necessity of working with the entire domain and no rationale is given for why just one of the two binding motifs was used for structural analysis.

2) It is no longer true that CaM uses the wrap-around mode of binding in the vast majority of cases.

3) Observing that Akt is not critical to calcium-induced activation of NHE1 is interesting, but this line of study would be much more significant if the kinase(s) that phosphorylate S648 were identified.

4) Subsection “Structure of the ternary complex of CaM and two NHE1 H1 helices”. The absence of NOEs between the two CaM domains/lobes suggests but does not “indicate” that the linker remains flexible. NMR relaxation, NMR RDCs, or SAXS analysis are required to draw any conclusion about inter-domain flexibility.

5) Discussion. This discussion should be modified because the based the authors' own data, which suggests that CaM will be pre-localized to NHE1 at basal levels of calcium.

6) Discussion. The reduction in the Kd value is quite modest. Hence, one cannot rule out that it is sufficient to pass the threshold required to alter CaM-NHE1 interaction in vivo or even in cells as detected by PLA.

---

## [Author Response]

[Editors’ note: the authors resubmitted a revised version of the paper for consideration. What follows is the authors’ response to the first round of review.]

Reviewer #1:The manuscript presents NMR, ITC and functional data for the protein calmodulin (CaM) and its interaction with NHE1. Importantly, the study suggests that CaM may bind to NHE1 in multiple ways and that CaM binding may stabilize dimerization of NHE1. The data and approach all seem satisfactory. However, I have a few serious concerns below about the interpretation of the NMR structure in Figure 4 (and implications in Figure 7), and the narrow scope of the approach, which all need to be addressed before a final evaluation can be made.1) The NMR structure of CaM bound to the NHE1 H1 helix (residues 620-650) in Figure 4 provides a very narrow view of the CaM interaction with NHE1, and is difficult to interpret the wider significance of this structure without knowing the overall stoichiometry of CaM binding to the full-length NHE1. Is the overall stoichiometry of CaM bound to full-length NHE1 known? If not, then it needs to be determined in this study in order to properly interpret the structure in Figure 4. For example, does the full-length NHE1 dimer bind to two CaM (2:2 complex) or just one CaM (1:2 complex)?

The in vivo stoichiometry of flNHE1 to CaM i is not known and in our opinion not trivial to obtain, since there is no experimental protocol established to express and purify dimeric flNHE1 or the complete cytosolic domain. Further, as we previously showed, flNHE1 and Calmodulin also interact indirectly via Calcineurin (Hendus-Altenburger et al., 2019), thus obtaining a reliable stoichiometry in vivo would require the separation of these interactions, which we do not consider feasible. We would be grateful for any suggestions of how to prove the stoichiometry in vivo. Equally importantly, the in vitro part of our study shows that there is not a single complex with ONE fixed stoichiometry but that there are various complexes formed depending on the conditions and availability of the binding partners. Thus, we propose that the stoichiometry of flNHE1 to CaM in vivo is similarly variable, allowing fine-tuning of NHE1 function in response to the specific subcellular environment, including the free concentrations of CaM and Ca^2+^ in conjunction with NHE1 phosphorylation state and other regulatory interactions.

If the CaM/NHE1 stoichiometry is 2:2, then the crystal structure of CaM bound to H1H2 complex (2YGG) already explains how CaM could stabilize dimerization of NHE1. The crystal structure (Figure 4—figure-supplement 1) shows that the CaM C-lobe binds to H1 of one molecule from NHE1 dimer, while the CaM N-lobe binds to H2 of the second molecule from NHE1 dimer. In other words, the two CaM lobes are domain swapped in the NHE1 dimer, which reveals how CaM stabilizes NHE1 dimerization as a 2:2 complex.

The crystal structure does not show a 2:2 complex and the article describing the crystal structure does not suggest a 2:2 complex in any way. The unit cell is 1:1 and the accompanying SAXS data is in clear disagreement with a 2:2 stoichiometry. The SAXS data is also in poor agreement with the proposed 1:1 complex (pdb-code: 2YGG), suggesting that the 1:1 model we propose in Figure 3d is actually the main complex in solution. The following are citations from the original paper describing the crystal-structur (pdb-code: 2YGG): “The overall structure shows a 1:1 binding stoichiometry of the NHE1CaMBR and CaM.”… “The scattering curves and the distance distribution function P(R) calculated from the crystal structure do not fit the experimental data perfectly (Figure 5A and B), where a shoulder at Dmax of 40–60 Å was more apparent for the complex structure. However, the deviations were small, and the agreement is fully consistent with a 1:1 stoichiometry of CaM and NHE1CaMBR in the complex. Any differences between the measured and calculated scattering curves can be attributed to flexibility of the NHE1CaMBR helix termini as well as the CaM helix connecting the lobes. The SAXS data show that the complex is monomeric in solution”.

So, if the overall CaM/NHE1 stoichiometry is measured to be 2:2, then the crystal structure already nicely explains how CaM stabilizes dimerization of NHE1 and the NMR structure in Figure 4 would merely confirm what was already known from the crystal structure. Alternatively, if the overall stoichiometry turns out to be 1 CaM per NHE1 dimer (1:2 complex), then the prediction from the crystal structure about domain swapping may be wrong, and the NMR structure in Figure 4 will in that case become far more interesting. For example, if the overall stoichiometry is 1:2, then the structure depicted in Figure 7f would explain how CaM stabilizes dimerization by having each CaM lobe bind to H1 (while H2 does not bind to CaM).

To our knowledge there is no data available that confirms that Ca^2+^/Calmodulin dependent activation of NHE1 is a one step process. So, it might very well be that complexes with different stoichiometries (1:1, 1:2, 2:2) are populated under different conditions in vivo, making the regulation of NHE1 a complex multistep process rather than a simple on/off switch.

This is at complete odds with the 2:2 structure (crystal structure) in which the CaM C-lobe binds to H1 and CaM N-lobe binds to H2.

All complexes described in this study in solution are at odds with the 1:1 (not 2:2) crystal structure. We think that crystallization is not the best method to study this dynamic, multi-state system and that the complex that crystallized in the previously published work is only populated to a very small extent in solution. In contrast, our solution NMR approach provides strong experimental evidence for the proposed models in Figure 3.

A simple way to distinguish the 1:2 model (Figure 7F) from the 2:2 model (crystal structure) would be to delete H2 from the full-length NHE1. The deletion of H2 should not affect CaM-induced dimerization of NHE1 in the 1:2 model, because CaM does not bind to H2 in this model. However, the deletion of H2 should abolish CaM-induced dimerization of NHE1 in the 2:2 model. So, in addition to measuring the overall CaM/NHE1 stoichiometry, it will also be necessary to prepare the H2 deletion construct of NHE1 and see if this deletion mutant impairs CaM-induced NHE1 dimerization.

We agree that many of the implications of our in vitro findings have to be put to a thorough test in vivo. And the suggested experiment is definitely an interesting option. But whatever the outcome of this experiment will be, it will be far from simple to interpret: first, the suggested interpretation is based on the assumption that the interaction has to be either a 1:2 or a 2:2 complex in the cell, neglecting that both forms of the complex might have a cellular function and regulate NHE1 differently and even be present simultaneously. Second, it will be difficult to pinpoint the functional effect of deleting the full H2 to the interaction with CaM, since other functional implications of this part of the protein might be present (for example the week self-association we also describe in the manuscript). Finally, as also pointed out in the manuscript, the dimerization of full-length NHE1 is supported by multiple interactions in both the transmembrane and cytosolic regions of the protein rather than only by H2. It would be exceedingly difficult, if not impossible, to dissect the contributions of each interaction from studies of the full-length protein in vivo and arrive at conclusive findings. Just as one example, Kd values for each interaction will be affected by the specific subcellular environment. Hence, while we fully acknowledge its limitations, a major strength of the in vitro experiments is the ability to study the interactions in a highly controlled environment.

2) Why was the NMR structure only reported for CaM/H1 complex? It would be far better to report the NMR structure of CaM bound to H1H2 or the entire C-terminal cytosolic domain. Does CaM bind to H1H2 (or cytosolic domain) as a 2:2 complex in solution? In other words, does CaM induce dimerization of H1H2 in solution (like what is implied in the crystal structure)? Does the C-terminal cytosolic domain form a dimer in solution, and does CaM binding stabilize this dimerization?

We stated in the manuscript: “It is, however, important to note that at 1:1 as well as 1:2 (CaM:H1H2), the NMR signals were drastically broadened compared to those of free CaM or of the 1:2 complex saturated with H1 (Figure 3A,D), indicating a dynamic equilibrium of states with different stoichiometries. Higher order complexes are also possible.” For this reason it was not possible for us to do a “classical” structure determination of the CaM:H1H2 complexes by NMR. In our opinion there is no way to determine the structure of this dynamic ensemble of complexes in a classic way since it is too small for cryo-EM, crystallization only gives an artifact of a minor subpopulation, and the dynamic nature of the complex in solution renders its NMR signals broad and inaccessible for higher dimensional NMR experiments needed for a structure determination. Our strategy of deducing structural information of the complex state with H1H2 by comparing the peak positions in the 15N-HSQC with the well-defined single state complexes of CaM with H1 or H2 is therefore the only way to come up with the models of the full complex at different stoichiometries.

3) Figure 3D shows a model of CaM bound to H1H2 in a 1:1 complex. This model predicts the CaM N-lobe binds to H1 and CaM C-lobe binds to H2, which is the exact opposite of what was shown in the crystal structure (CaM N-lobe binds to H2). The discrepancy here needs more explanation. How do you know that the CaM N-lobe is bound to H1 in the 1:1 complex here? The NMR structure of CaM bound to H1H2 is needed here to prove whether the N-lobe actually binds to H1 or H2 in the CaM/H1H2 complex.

At a 1:1 stoichiometry of CaM:H1H2 all 70 peaks (140 chemical shifts) of the calmodulin N-lobe overlap strikingly well with the state where CaM is bound to H1 and the same is true for the 70 peaks of the C-lobe and H2 (Figure3D-F). Thus, the 2 models we propose in Figure 3A,D are very well supported by experimental evidence (~280 observations of chemical shifts on calmodulin that are additionally backed up from the NMR studies on labelled NHE1 as shown in the figure supplement 2), so we are not sure what more evidence we can provide, since a full structure determination is not possible for the reasons described above.

4) Figure 7 shows various types of CaM/NHE1 interaction. Unfortunately, the most interesting interaction (2:2 complex predicted from the crystal structure) is not shown. An extra panel should be added in Figure 7 to show the 2:2 interaction (implied from the crystal structure), in which the CaM N-lobe binds to H2 from one molecule of NHE1 dimer, and the CaM C-lobe binds to H1 from the other molecule of NHE1 dimer. The 2:2 interaction should depict two CaM molecules interacting in a crisscrossed fashion with the NHE1 dimer. Again, it will be important here to experimentally test the 1:2 model (Figure 7F) versus the 2:2 model implied by the crystal structure. The H2 deletion described above in point 1 can distinguish these models. The most direct way to determine which model is functionally most relevant would be to measure the overall stoichiometry of CaM/NHE1. Is it 1:2 or 2:2 or perhaps a composite of the two? Without knowing the CaM/NHE1 stoichiometry, the functional relevance of the NMR structure in Figure 4 cannot be properly understood.

As explained above, this is based on a misunderstanding, as there is *no* experimental evidence for the existence of a 2:2 complex, neither in our submitted study nor in the study that solved the crystal structure. The spatial proximity of two cytosolic tails in a flNHE1 dimer makes this a feasible alternative organization, as we mention in the discussion part of the paper. Nevertheless, we wanted to only include models in our final figure where we have solid experimental evidence for.

Reviewer #2:1) There is an obvious gap in logic that is not addressed, i.e. why the solution NMR structure was determined for only half of the NHE1 interaction domain. The ITC data clearly point to the necessity of working with the entire domain and no rationale is given for why just one of the two binding motifs was used for structural analysis.

Thank you for this comment, which likely reflects that we have not sufficiently clearly explained the basis for our strategy:

We state in the manuscript: “It is, however, important to note that at 1:1 as well as 1:2 (CaM:H1H2), the NMR signals were drastically broadened compared to those of free CaM or of the 1:2 complex saturated with H1 (Figure 3A,D), indicating a dynamic equilibrium of states with different stoichiometries. Higher order complexes are also possible.” For this reason, it is not possible to do a “classical” structure determination of the CaM:H1H2 complexes by NMR. In our opinion there is no way to determine the structure of this dynamic ensemble of complexes in a classical way since it is too small for cryo-EM, crystallization only provides insight to a minor subpopulation, and the dynamic nature of the complex in solution renders its NMR signals broad and inaccessible for higher dimensional NMR experiments needed for a structure determination.

Our strategy of deducing structural information of the complex ensemble of CaM with NHE1-H1H2 was therefore to compare the peak positions in the ^15^N-HSQCs with the well-defined single state complexes of CaM with either H1 or H2. We believe this is the only way to deduce models of the full complex at different stoichiometries. Indeed, the state populated with excess H1 is the same state we see when CaM is saturated with H1H2. The NMR spectra overlap nicely and thus allow access to the CaM:H1H2 structure through the CaM:H1 complex.

At a 1:1 stoichiometry of CaM:H1H2 all 70 peaks (140 chemical shifts) of the CaM N-lobe (or N-domain) overlap strikingly well with the state where CaM is bound to H1. The same is true for the 70 peaks of the C-lobe (or C-domain) and H2 (Figure 3D-F). At a 1:2 stoichiometry all 140 peaks of CaM:H1H2 overlap with the state where CaM is bound to H1. This indicates the NMR-structure we solved is equivalent to the major state in solution at a stoichiometry of 1:2 (CaM:H1H2). So the two models we propose in Figure 3A,D are very well supported by experimental evidence: ~280 observations of chemical shifts on CaM that are additionally backed up from the NMR studies on labelled NHE1 as shown in Figure supplement 2 and the size determinations in our revised manuscript. Since a direct structure determination is not possible for the reasons described above, we believe our approach is the most appropriate, and we think we can provide ample experimental evidence for the conclusions that we draw.

While this was mentioned in the original manuscript and the data shown, we agree that we did not make this point sufficiently clear. To emphasize this important information, we have therefore rewritten the manuscript, and now initiate this section by presenting the CaM:H1H2 complex and highlighting that the quality of these spectra did not allow for its full structure determination. We then explain our comparative strategy for elucidating the structure of the 1:2 complex using the complexes of CaM with the individual peptides. This has resulted in a complete rewrite of the text in this part of the manuscript, and we now explain the logics of our approach much clearer (please see the revised manuscript). We thank the reviewer for spurring this rewrite, which we think have improved the clarity of our strategy.

2) It is no longer true that CaM uses the wrap-around mode of binding in the vast majority of cases.

Thank you for this comment. We agree, and the statement has been rephrased to “In most cases where this has been studied to date, CaM wraps both its lobes around its α-helix-forming targets, exemplified by the interaction with myosin light chain kinase (Ikura et al., 1992; Meador, Means and Quiocho, 1992). However, it is now clear that a single CaM binding mode or a single binding motif does not exist.”.

3) Observing that Akt is not critical to calcium-induced activation of NHE1 is interesting, but this line of study would be much more significant if the kinase(s) that phosphorylate S648 were identified.

We agree that this mechanism would be highly interesting to unravel. We therefore carried out a series of new experiments. As our data showed that both Akt and protein kinase C-δ (PKCδ) phosphorylated this residue in vitro (Figure supplement 9B), we tested the effect of two well-established PKC inhibitors, bisindoylmaleimide I (BIM 1) – targeting PKCα, -β1, -β2, -γ, -δ, and –ε, and Gö6983 – targeting PKCα, PKCβ, -γ, -δ and -ζ. The data, presented as new Figure 6f, show that preincubation with even high doses of these PKC inhibitors does not prevent ionomycin-induced NHE1 activation. Thus, we conclude that while Akt and PKC can phosphorylate S648, they are not essential for calcium-induced NHE1 activation. It may be noted that BIM I also inhibits GSK3 (Hers, Tavare and Denton, 1999), hence, this kinase, which has a similar consensus sequence as Akt and PKC, is also not required for the Ca^2+^-induced activation. Given that we identified several other potential phosphosites on NHE1 that may play a role but require extensive further studies, a full understanding of the mechanism will await further studies. The new data have been presented on p 17-18 of the revised manuscript.

4) Subsection “Structure of the ternary complex of CaM and two NHE1 H1 helices”. The absence of NOEs between the two CaM domains/lobes suggests but does not “indicate” that the linker remains flexible. NMR relaxation, NMR RDCs, or SAXS analysis are required to draw any conclusion about inter-domain flexibility.

We fully agree and have changed *indicate* to *suggest* in the revised manuscript. Thank you for pointing this out.

5) Discussion. This discussion should be modified because the based the authors' own data, which suggests that CaM will be pre-localized to NHE1 at basal levels of calcium.

Thank you for pointing this out. We agree and have rephrased the paragraph in the revised manuscript to read:

“As CaM is limiting in a cellular context, the two latter scenarios will be modulated both by local [Ca^2+^]_i_ and by CaM interactions with other proteins at any given time. Although the total cellular [CaM] is 5-6 µM as measured in various cell types, the global free Ca^2+^-CaM concentration in a cell has been estimated to around 50 nM, i.e. ~1% of total [CaM] (Persechini and Stemmer, 2002; Saucerman and Bers, 2012). Both [CaM] and [Ca^2+^]_i_ may, however, reach much higher concentrations locally, e.g. close to Ca^2+^ channels (Mori, Erickson andYue, 2004), and the precise local Ca^2+^-CaM availability around NHE1 will be dependent on local and global Ca^2+^ signaling and competition with other CaM binding proteins of various affinities. Our data suggest that in a cellular context, CaM is bound to NHE1 at basal global [Ca^2+^]_i_. It is however possible that the local [Ca^2+^]_i_ immediately surrounding the NHE1 C-tail may reach substantially higher levels due to local complex formation with Na^+^/Ca^2+^ exchanger 1(NCX1) (Yi et al., 2009). Hence, the precise Ca^2+^ requirement for binding in a cellular context may be higher than the global [Ca^2+^]_i_.”

6) Discussion. The reduction in the Kd value is quite modest. Hence, one cannot rule out that it is sufficient to pass the threshold required to alter CaM-NHE1 interaction in vivo or even in cells as detected by PLA.

While the reduction is actually a factor of 10 which is not in our view insignificant, we do agree that such a scenario is feasible in vivo, especially in a high-competition scenario. We have thus rephrased the text to “….proximity to CaM in cells. This may reflect that the effect of phosphorylation on the interaction was not strong enough to prevent interaction in vivo. Interestingly, the observation mirrors….”.